# Data-driven design of molecular nanomagnets

Yan Duan[1,2,5], Lorena E. Rosaleny [1,5] ✉, Joana T. Coutinho [1,3,5] ✉,
Silvia Giménez-Santamarina[1], Allen Scheie [4], José J. Baldoví[1],
Salvador Cardona-Serra[1] & Alejandro Gaita-Ariño [1] ✉

Three decades of research in molecular nanomagnets have raised their magnetic memories from liquid helium to liquid nitrogen temperature thanks to a wise choice of the magnetic ion and coordination environment. Still, serendipity and chemical intuition played a main role. In order to establish a powerful framework for statistically driven chemical design, here we collected chemical and physical data for lanthanide-based nanomagnets, catalogued over 1400 published experiments, developed an interactive dashboard (SIMDAVIS) to visualise the dataset, and applied inferential statistical analysis. Our analysis shows that the Arrhenius energy barrier correlates unexpectedly well with the magnetic memory. Furthermore, as both Orbach and Raman processes can be affected by vibronic coupling, chemical design of the coordination scheme may be used to reduce the relaxation rates. Indeed, only bis-phthalocyaninato sandwiches and metallocenes, with rigid ligands, consistently present magnetic memory up to high temperature. Analysing magnetostructural correlations, we offer promising strategies for improvement, in particular for the preparation of pentagonal bipyramids, where even softer complexes are protected against molecular vibrations.

Molecular nanomagnets were reported for the first time at the beginning of the 1990s, when $Mn_{12}O_{12}(CH_3COO)_{16}(H_2O)_4$ was discovered to display magnetic hysteresis in analogy to classical magnets, but with a quantum tunnelling mechanism for the relaxation of the magnetisation[1,2]. This polynuclear magnetic complex was the first of a plethora of single-molecule magnets (SMMs). The term was coined for systems behaving as hard bulk magnets below a certain temperature, but where the slow relaxation of the magnetisation is of purely unimolecular origin. Their magnetic behaviour can be approximated to that of an effective anisotropic magnetic moment arising from the exchange interactions between the spins of the metal ions. The reversal of this giant anisotropic spin occurs by populating excited spin states and overcoming an energy barrier. Hence, the thermal dependence of the relaxation rate was described by the Arrhenius equation (Fig. 1b), using this effective energy barrier ($U_{eff}$) and a pre-exponential factor ($\tau_0$)[3]. Both parameters were not extracted directly from the hysteresis loop (see Fig. 1c), but rather from the combined frequency- and temperature-dependence of the so-called out-of-phase component of the ac susceptibility ($\chi''$, see Fig. 1a)[3,4]. Given sufficient experimental information, considering other processes in the fit, such as the Raman process and Quantum Tunnelling of the Magnetisation (QTM), in principle results in more accurate values, which are denoted as $U_{eff,ff}$, $\tau_{0,ff}$. It also allows extracting additional parameters $C$, $n$ to characterise Raman, and $\tau_{QTM}$ (Fig. 1b).

[1]Instituto de Ciencia Molecular (ICMol), Universitat de València, C/Catedrático José Beltrán 2, 46980 Paterna, Spain. [2]Spin-X Institute, South China University of Technology, 510641 Guangzhou, People's Republic of China. [3]Centre for Rapid and Sustainable Product Development, Polytechnic of Leiria, 2430-028 Marinha Grande, Portugal. [4]Neutron Scattering Division, Oak Ridge National Laboratory, Oak Ridge, TN 37831, USA. [5]These authors contributed equally: Yan Duan, Lorena E. Rosaleny, Joana T. Coutinho. ✉e-mail: rosaleny@uv.es; Joana.t.coutinho@ipleiria.pt; gaita@uv.es

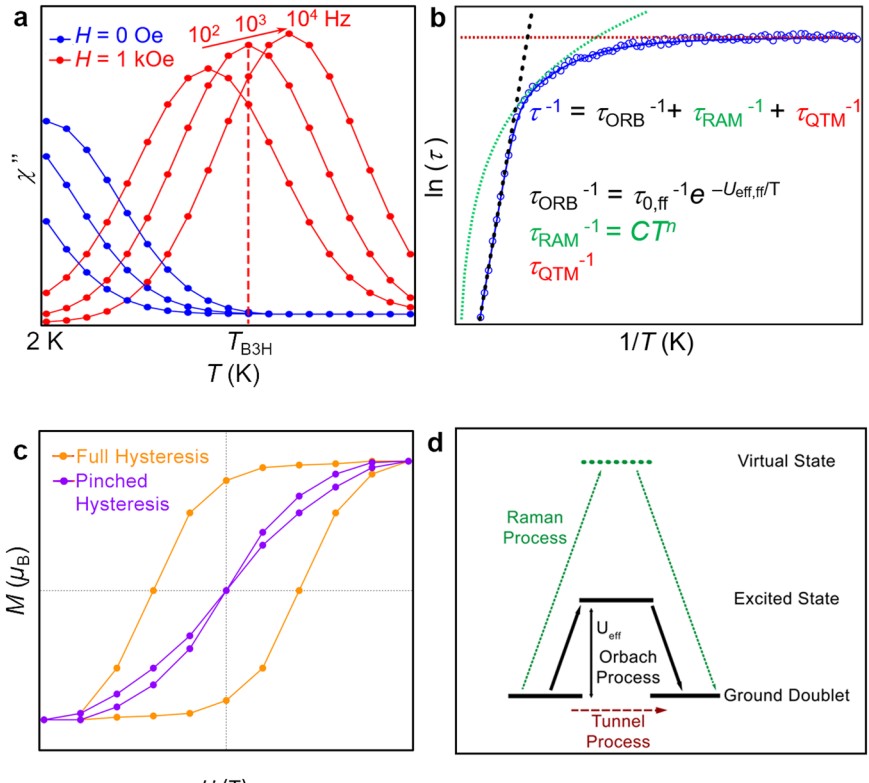

**Fig. 1 | Main magnetic concepts employed in this study.** Slow relaxation of the magnetisation in SIMs can manifest in different ways. **a** Spin blocking is often characterised by a temperature- and frequency-dependent out-of-phase ac susceptibility $\chi''$, being $T_{B3H}$ the temperature for the maximum $\chi''$ at $10^3$ Hz in the presence of a magnetic field. **b** These relaxation dynamics have most often been modelled as an Orbach process (black dots), using the Arrhenius equation. Raman (green dots) and quantum tunnelling (red dots) processes can also be relevant. **c** Magnetic hysteresis can be full (orange) or "pinched", also known as butterfly (purple), where the latter signals a fast relaxation at zero magnetic field. **d** Scheme of different relaxation processes: tunnelling involves just the states within the ground doublet, Orbach process takes place via an excited state, and Raman process happens via a virtual state.

The best metric for slow relaxation is hysteresis temperature ($T_{hyst}$), the highest temperature at which the system presents magnetic bistability. This translates in open hysteresis as a necessary condition for achieving molecular magnets. The first SMMs exhibited low values of $T_{hyst}$, which was attributed to their modest effective energy barrier ($U_{eff} \approx 50$ K)[5]. Initial models based on effective spin Hamiltonians gave rise to the relation $U_{eff} = DS_z^2$ and concluded that the best strategy to raise $U_{eff}$ and, therefore, to improve the maximum $T_{hyst}$ is to maximise the total effective spin ($S$), rather than the magnetic anisotropy ($D$)[6]. Indeed, the latter is a less straightforward target for the synthetic chemist[7]. Despite great effort toward the synthesis of such systems and an abundance of molecules with increasing values of $S$, very little progress was made in the first decade in terms of increasing $U_{eff}$ or $T_{hyst}$[8].

In the 2000s, a novel class of molecular nanomagnets emerged, namely bis-phthalocyaninato (Pc) double deckers[9]. This second generation of SMMs, commonly known as Single Ion Magnets (SIMs), is based on mononuclear complexes containing a single magnetic ion embedded in a coordination environment. They constitute the smallest molecule-based magnet and their properties arise from strong spin–orbit coupling which, combined with the crystal-field interaction with the surrounding ligands, results in an enhanced magnetic anisotropy when compared to SMMs. Identical data treatment using the Arrhenius equation led to effective energy barriers $U_{eff}$ up to an order of magnitude higher for SIMs based on rare-earth ions when compared to those of polynuclear metal complexes of the $d$-block. The characteristic maxima in the out-of-phase component of the ac susceptibility $\chi''$ also moved to higher temperatures (Fig. 1a), but $T_{hyst}$ did not increase as significantly.

After the germinal LnPc$_2$, different chemical families such as polyoxometalates[10] and metallocenes[11] were shown to exhibit slow relaxation of the magnetisation of purely molecular origin (Fig. 2). Initially, $Tb^{3+}$ and $Dy^{3+}$ ions were the most commonly studied. These present an equatorially expanded $f$-electron charge cloud and are known as oblate (see Supplementary Section 1). Success cases were also found for lanthanide ions with axially elongated $f$-electron charge cloud (prolate ions, e.g., $Er^{3+}$, $Tm^{3+}$, $Yb^{3+}$). The realisation that lanthanide SIMs were not restricted to a single chemical strategy inspired a large community of chemists. As a result, between 2003 and 2019 SIM behaviour was reported in over 600 compounds, and above a third of these compounds actually displayed magnetic hysteresis. No single chemical strategy has dominated in terms of reported examples, although many approaches have been paradigmatic (e.g., the use of radicals[12,13] and diketonates[14]). Recent efforts have been made to offer some perspective[15–18]. Nevertheless, anecdotal claims from proven strategies are hard to distinguish, as so many studies pursuing independent approaches have been reported. Modern techniques of data analysis and visualisation can contribute to remedy this knowledge gap. In particular, dashboards are intuitive graphical applications for dynamic data visualisation and information management, of growing popularity in different fields[19–21].

The present work firstly aims to rationalise the correlations among the different physical variables involved in SIMs. A common hypothesis is that the parameters arising from the ac susceptometry (e.g., $U_{eff}$) are well correlated with the experimental values (e.g., $T_{hyst}$). This, however, has actually been challenged in various ways[18,22,23]. Over the years, various theoretical approaches have put the focus on the rationalisation of different physical processes and parameters[22–25].

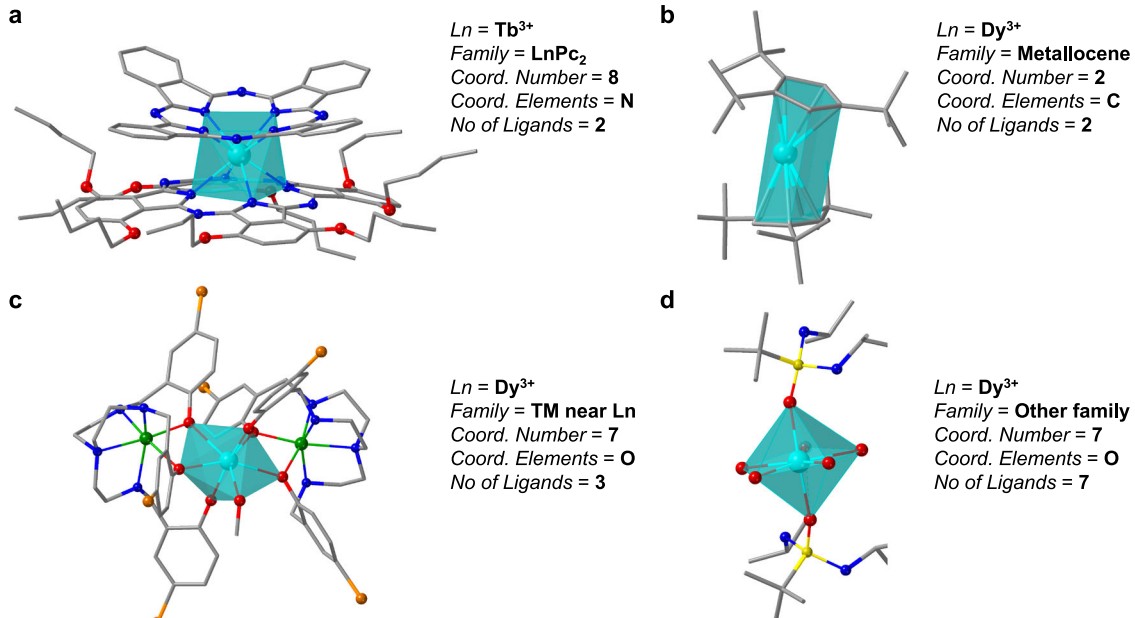

Fig. 2 | Molecular structures of some representative lanthanide-based SIMs from different chemical strategies and some of their chemical descriptors. **a** Pc double deckers, abbreviated as LnPc$_2$ ($T_{hyst}$ = 2 K)[72]. **b** Metallocene complex LnCp*$_2$ ($T_{hyst}$ = 60 K)[42], see Supplementary Information 1 for our CN criterion for the metallocene family. **c** A complex with the introduction of a diamagnetic TM ion near the lanthanide ion, [Zn$_2$DyL$_2$(MeOH)]⁻ (L is a tripodal ligand, 2,2′,2″-

(((nitrilotris(ethane-2,1-diyl))tris(azanediyl))tris(methylene))tris-(4-bromophenol)) ($T_{hyst}$ = 11 K)[73]. **d** [L$_2$Dy(H$_2$O)$_5$]³⁺ (L = ᵗBuPO(NH$^i$Pr)$_2$), a complex outside the main categories of the present study, which was classified as other families ($T_{hyst}$ = 30 K)[46]. (colour scheme for atoms: green, Zn; cyan, Tb or Dy; grey, C; blue, N; yellow, P; orange, Br; red, O. Hydrogen atoms are not shown for clarity).

Secondly, in order to provide the synthetic efforts with a data-driven chemical design guide, we apply the techniques of third generation computational chemistry[26]. Here, we started by collecting a high-quality dataset from published data and represent it in an interactive dashboard. Then, we statistically analysed the correlations between molecular descriptors and physical parameters. As a second phase of the work, we expanded this dataset to rationalise the correlations found in the first phase, and to analyse the influence of the shapes of coordination environments on the magnetic dynamics.

## Results

### A dataset and interactive dashboard for lanthanide SIMs
We built a dataset of the most relevant chemical and physical properties of 1411 lanthanide SIM samples collected from 448 scientific articles (Supplementary file) published between 2003 and 2019 and developed a user-friendly dashboard-style web application named SIMDAVIS (Single Ion Magnet DAta VISualisation) to host it. The dataset contains over 10000 independent pieces of chemical information, as well as over 5000 independent pieces of physical (magnetic) information. Furthermore, the dataset is hierarchically clustered into magnetostructural taxonomies (see Supplementary Sections 4 and 6) in order to pave the way for further analysis, including machine-learning studies, a field that is now on the rise[27,28]. Indeed, data taxonomies are powerful tools to make sense of data, since they provide ordered representations of the formal structure of knowledge classes or types of objects within a data domain[29].

Each chemical family that has been widely explored in this field is claimed to be promising as molecular nanomagnet, usually by citing the best reported case. However, it is crucial to avoid getting distracted by the occasional well-behaving example and instead to evaluate the general behaviour of each chemical strategy. Do members of a family generally present a slow relaxation of the magnetisation, in terms of ac susceptometry and/or hysteresis? To evaluate this against a common reference, our dataset allows comparing the overall performance of samples in each family with the performance of the "Mixed

Ligands" and "Others" Families, that act here as a sort of control group. Similarly, since Tb³⁺ and Dy³⁺ ions are oblate, as well as the cases where the record results have been obtained, it is also commonly assumed that in general complexes with oblate ions result in better SIM properties compared with prolates. Our dataset should be able to test this.

The question remains of what does one mean by better SIM properties, or, as eloquently put recently, "How do you measure a magnet?"[30] Blocking temperature definitions in recent works include the temperature at which the relaxation time is 100 s ($T_{B2}$), the temperature at which there is a maximum in the zero-field cooled susceptibility ($T_{B1}$), and the maximum temperature at which hysteresis is observed ($T_{hyst}$). Unfortunately, only a small part of the articles provided any of these parameters, whereas older bibliography favoured only $T_{hyst}$. This potentially introduces a severe publication bias that our dataset tackles by including information about ac susceptometry, $U_{eff}$ and hysteresis (see Supplementary Section 1.1).

Qualitative and quantitative information based on the almost ubiquitous ac susceptibility measurements was invaluable for our analysis. Since there is an ac curve at (or near) 1000 Hz frequency in virtually all works in the field (given that both MPMS and PPMS magnetometers cover this range), we chose the maximum of the out-of-phase ac curve at this frequency as the basis for our most abundant qualitative and quantitative data source. At a given frequency, $\chi''$ does not necessarily present a maximum; an external magnetic field facilitates this effect by cancelling QTM. In this study, we register the temperature of this maximum as $T_{B3}$ ($T_{B3H}$), the blocking temperature at $10^3$ Hz in absence (in presence) of a magnetic field (Fig. 1a).

$U_{eff}$ is also very widely employed and assumed to be a good descriptor of magnetic behaviour. In contrast, in Arrhenius-type fits generally little attention is paid to $\tau_0$; according to a simplified two-phonon Orbach model, the two variables are supposed to correlate[31]. At the same time, $U_{eff}$ is rightfully criticised as an oversimplification that overlooks physically independent mechanisms (notably, Raman) that could be dominating the behaviour (Fig. 1d). Our dataset aimed to answer these questions.

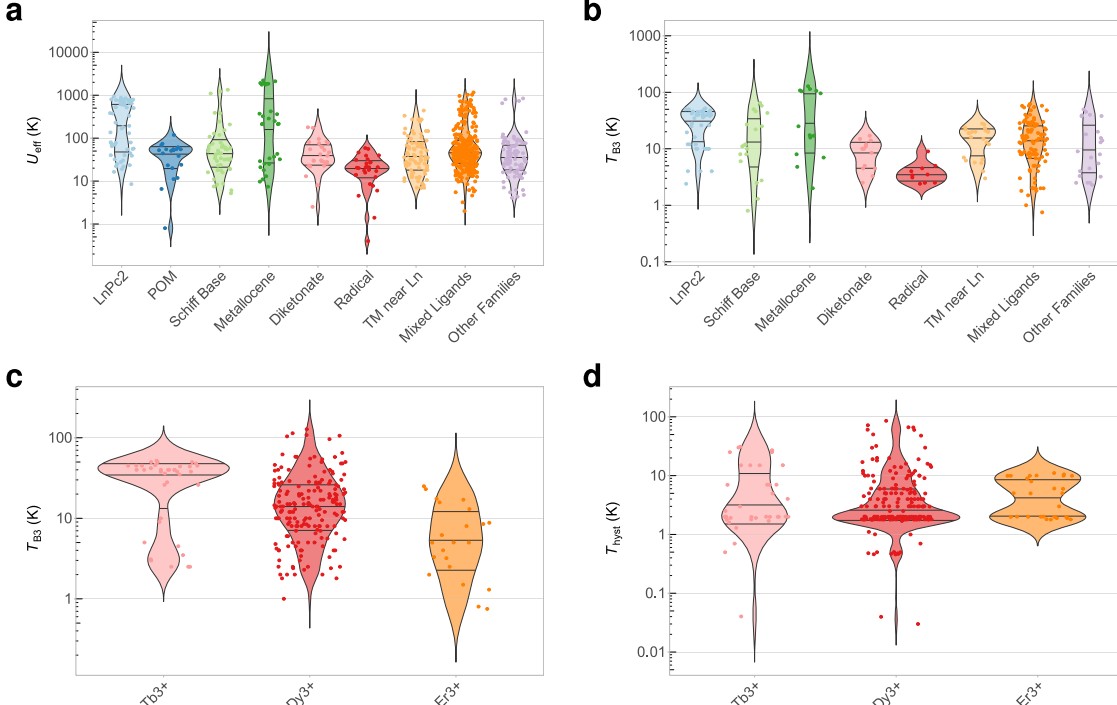

**Fig. 3 | Violin plots and bar charts relating magnetic relaxation behaviour with the main chemical parameters.** The width of each violin plot is proportional to the density of data points for this range of values, and the horizontal stripes mark the quartiles. **a** Values of $U_{eff}$ for samples in each chemical family. **b** Values of $T_{B3}$ for samples in each chemical family. **c** Values of $T_{B3}$ for samples containing $Tb^{3+}$, $Dy^{3+}$, $Er^{3+}$. **d** Values of $T_{hyst}$ for samples containing $Tb^{3+}$, $Dy^{3+}$, $Er^{3+}$.

Finally, note that a remnant magnetisation at $H = 0$ is a defining feature for molecular nanomagnets. If remanence is lacking, it is not feasible to store long-term information on a molecule. One can ask: is a molecule that shows out of phase magnetic susceptibility as a response to alternating current but no hysteresis, really a molecular nanomagnet? There is, however, a link between short- and long-term magnetic memory, so we included the wider definition of SIMs in the present study, as detailed above.

SIMDAVIS allows the chemical community to visualise the key relationships between chemical structures and physical properties in our catalogue of SIMs. Our interactive dashboard can be directly invoked by accessing the internet site where it is located[32]. It is organised in 6 main tabs: Home, ScatterPlots, BoxPlots, BarCharts, Data (View Data and Download Data) and About SIMDAVIS (Variables, Authors, Feedback&Bugs, Changelog and License) as we can observe in Supplementary Fig. 11.

In the SIMDAVIS dashboard, the most versatile source of graphical information is the "ScatterPlots" tab, where an example plot is explained in Supplementary Fig. 11. The next two tabs display the data in complementary ways. The "BoxPlots" tab allows to examine the distribution of each SIMs quantitative property vs. a categorisation criterion, e.g., we can see the distribution of $U_{eff}$ values as a function of the coordination elements. The boxplot for each category is shown, including the median and the interquartile range. The "BarCharts" tab allows the exploration of the frequency of different qualitative variables in our dataset. Stacked bar graphs allow the simultaneous analysis of two qualitative variables, e.g., we can display, for each chemical family, the number of samples, which present magnetic hysteresis. The "Data" tab is a powerful interface to browse the dataset, featuring the possibility to choose the data columns to show, ordering in ascending or descending order, and filtering by arbitrary keywords; it also permits downloading all data, including links to the CIF files, when available. Finally, the "About SIMDAVIS" tab contains information about the variables contained in the dataset.

## Statistically driven chemical design of SIMs

SIMDAVIS allows the visualisation of the relationships between chemical and physical variables in SIMs, and thereby enables determining the main variables that the synthetic chemist needs to consider to obtain the desired physical properties. We will first analyse this qualitatively employing a series of boxplots, violin plots and bar charts (see Fig. 3 and Supplementary Figs. 11.1–11.6, 12.1–12.5, 13.1–13.2). The full statistical analysis is presented in Supplementary Sections 4, 5 and 6.

First, let us focus on the effective energy barrier $U_{eff}$ and the blocking temperature $T_{B3}$ (the temperature for maximum out-of-phase ac susceptibility $\chi''$ at $10^3$ Hz, see Fig. 1). From Fig. 3 and Supplementary Figs. 11.1–11.4, we can see that the chemical families with a distinctly good behaviour are $LnPc_2$ and metallocenes, with median values of $U_{eff} > 200$ K and $T_{B3} > 30$ K. Equivalently, one can see that $Dy^{3+}$ and $Tb^{3+}$ present somewhat higher $U_{eff}$, $T_{B3}$ than the rest (Supplementary Figs. 11.3c, 11.4a) and that, in general, oblate ions perform better than prolate ions, for both properties. In addition, non-Kramers ions present higher median $T_{B3}$ but similar $U_{eff}$ values compared with Kramers ions.

Now, let us analyse the maximum hysteresis temperature $T_{hyst}$. The only chemical family with a distinct positive behaviour is the metallocene family. More surprisingly, $Er^{3+}$ complexes have distinctly high hysteresis temperatures, markedly with a higher median than $Dy^{3+}$ or $Tb^{3+}$ complexes. This is in sharp contrast with their relative $T_{B3}$ values, which are consistently much lower in the case of $Er^{3+}$ complexes. This not only indicates that searching for equatorial environments, precisely the ones that favour good magnetic properties in $Er^{3+}$ complexes[24], often results in more rigid ligands, but also indicates an underexplored territory. It is certainly possible that chemical modifications of $[Er(COT)_2]^-$ (or other $Er^{3+}$ record-bearing complexes) designed to optimise the detrimental effect of molecular vibrations may achieve records that are competitive with $DyCp^*_2$. Prolate ions are consistently—and surprisingly—better than the oblate ones, having a higher median value for $T_{hyst}$. This is again in contrast with the

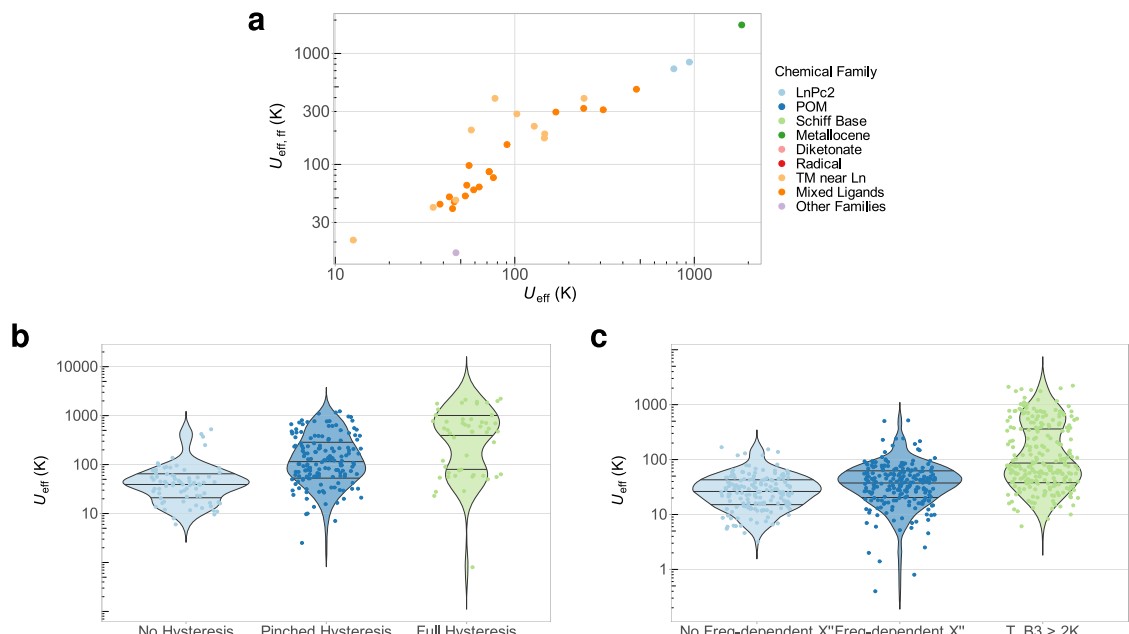

**Fig. 4 | Main dependencies between the physical variables. a** Dependence between $U_{eff}$ and $U_{eff,ff}$. **b** Distribution of $U_{eff}$ for samples depending on their qualitative hysteresis behaviour, **c** Distribution of $U_{eff}$ for samples depending on their qualitative ac $\chi''$ susceptibility behaviour. For a more complete analysis, see Supplementary Sections 5.1 and 6.

opposite behaviour that is observed for $T_{B3}$ and $U_{eff}$, and possibly again due to the influence of $Er^{3+}$ complexes with their more rigid equatorial environments. This behaviour of $Er^{3+}$ is unexpected after a recent theoretical contribution[33], which calculated the electronic structure of $Er[N(SiMe_3)_2]_3$ variants, concluding that no geometrical optimisation can significantly improve $U_{eff}$ for $Er^{3+}$. Nevertheless, all the high $U_{eff}$ cases involving $Er^{3+}$ in our database are based in the COT ligand, meaning our differing conclusions stem from different chemical strategies.

Finally, the coordination number and the number of ligands do have an influence on the statistically expected hysteresis temperature, with the best ones being 2 and 7 in the case of the coordination number and just 7 for the number of ligands. As we will discuss below, there are chemical insights to be gained from this if one analyses the influence of the coordination environment shape.

To put all these trends into perspective, it is important to numerically analyse the connection between the different variables and the clustering of our data. A lognormal analysis (see Supplementary Section 4.3) shows that the three main chemical variables, namely the chemical family, the lanthanide ion and the coordination elements, are sufficient to reasonably explain the variation of values of the others. This means that there is a limit on the information one can independently extract from the rest of the chemical variables. Multiple correspondence analysis (see Supplementary Sections 4.1, 4.2) suggests a chemical clustering that consists in three small groups, namely $Gd^{3+}$ complexes, metallocenes and $LnPc_2$ double deckers, and two much larger groups with a large overlap with oblate and prolate ions respectively. A factorial analysis of mixed data considering also all magnetic information available (see Supplementary Section 6) simplifies the clustering to three groups. Again, the two distinct families present a large overlap with metallocenes and $LnPc_2$ double decker chemical families, both of them presenting significantly better properties than the other kinds of samples. Finer clustering categorisations are possible and indeed available in the dataset. These taxonomies can serve to guide future theoretical work. In the current stage, they mainly serve to confirm that, in layman's terms, all chemical families within our current dataset present basically indistinguishable magnetic behaviours, except metallocenes and double deckers.

Further insight is provided representing the reported behaviour of magnetic hysteresis and *ac* magnetic susceptibility as a function of different chemical variables, namely (i) chemical family, (ii) metal ion, (iii) coordination number and (iv) coordination elements (Fig. 3, Supplementary Figs. 12.1 to 12.5). Remember that for hysteresis we are limited by the minority of the samples where hysteresis or its absence is reported; in the vast majority of the cases this information is lacking. Nevertheless, it is apparent that certain chemical families such as $LnPc_2$ and metallocenes tend to display hysteresis, with diketonates being in a distant third position. In contrast, complexes based on POMs or on Schiff bases seldomly report hysteresis, and actually tend to not even present out of phase *ac* signals (Supplementary Fig. 12.2). In terms of metal ions, $Dy^{3+}$, $Tb^{3+}$ and $Er^{3+}$ are clearly the best behaved (Fig. 3 and Supplementary Fig. 12.3).

The suggestion of future directions to guide synthetic efforts requires a more detailed study of the physical mechanisms of relaxation and the predictive power of the different parameters, as well as taking into account the shape of the coordination environment. This information is not immediately available from the literature. We performed these tasks and here we present the results in the following sections.

## Orbach mechanism: oversimplified, predictive… a function of vibronic coupling?

A key question is: how much have the analyses in this field been affected by the simplified assumption that SIMs relax via an Orbach mechanism, characterised by $\tau_0$ and $U_{eff}$? We strived to quantify up to what level the value of $U_{eff}$ is well correlated with the slow relaxation of the magnetisation. Ultimately, we wanted to verify whether a true correlation with magnetic relaxation is only possible when available data allows the fitting of $U_{eff,ff}$. A visual inspection (Fig. 4a), a categorical analysis (Fig. 4b, c), an in-depth statistical analysis of all physical parameters based on the Akaike Information Criterion (Supplementary Section 5.3) and a factorial analysis of mixed data (Supplementary Section 6) conclude that $U_{eff}$ derived from a simple Arrhenius plot is,

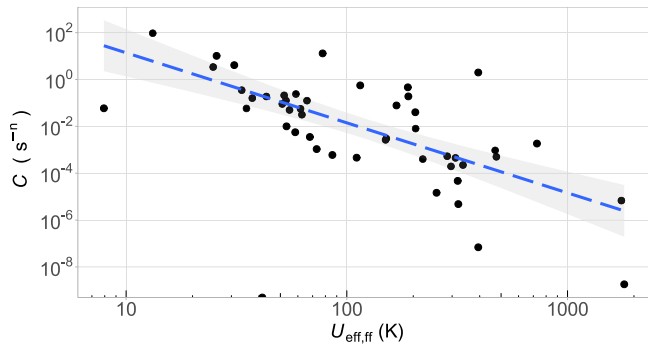

**Fig. 5 | Relation between $U_{eff,ff}$ and Raman relaxation parameter $C$, in logarithmic form.** Data points are accompanied by a least square fit including confidence intervals.

by itself, an excellent predictor for magnetic behaviour. Supplementary Section 5.4 presents the full discussion of this question. As an alternate approach to evaluate the validity of the Orbach mechanism, it has been pointed out that frequently, as $U_{eff}$ increases, $\tau_0$ decreases, leaving relaxation times essentially constant[5]. An approximate relation between $\tau_0$ and $U_{eff}$ can be derived for a two-phonon Orbach process within a Debye phonon model[31]. Fitting experimental data to it results in a large dispersion but only moderate deviations from the expected parameter range, meaning these approximations can be useful (Supplementary Section 5.5 for details)[30].

Having shown that $U_{eff}$ is very successful—perhaps unreasonably so—at predicting the magnetic behaviour, we turn our attention to the other relaxation pathways, notably Raman and quantum tunnelling of the magnetisation processes. The former is characterised by a prefactor $C$ and an exponent $n$, whereas the latter by a time $\tau_{QTM}$. Since fits including this information are relatively scarce, one needs to note that this phase of the analysis has much less statistical power. We extracted $C$, $n$, $\tau_{QTM}$ from all samples which presented $U_{eff,ff}$, and explored possible correlations among the different parameters, in logarithmic form. We found that $\log(U_{eff})$ seems to correlate quite well with $\log(C)$ (see Supplementary Fig. 34, top), and this correlation is perhaps more clear with $\log(U_{eff,ff})$ (see Fig. 5). In particular, cases with high $U_{eff,ff}$ (>200 K) present a low $C$ (<$10^{-3}$ s$^{-1}$) and vice versa. While the number of points is limited, this correlation is statistically sound (see Supplementary Section 9.3). However, these are supposed to be two fully independent mechanisms, so what could be the reason behind this apparent coincidence?

First we need to be aware of the fact that even a fit considering different relaxation mechanisms is a simplification. Indeed it has been shown that the anomalous Raman exponents so often found in these fits come from this fact[34]. These fits assume that one is studying an experiment with 3 physical processes that can be independently parameterised. Instead, many more processes are simultaneously taking place, including alternate multi-phonon Orbach mechanisms, competing Raman mechanisms dominated by different vibrational frequencies and the direct process. This means it is not surprising that, when fitting an overly complex process with a few parameters, some of them are unphysically correlated.

However, there is also a possible physical reason behind the correlation we found. It is the coupling between spin states and vibrations: a recently recognised key parameter in molecular spin dynamics[35–37]. Spin-vibration coupling is a common factor for both relaxation pathways[38], and plays a vital role in both Raman[37,39] and Orbach[37,40,41] mechanisms. This means that a strong crystal field is not a sufficient condition for a high $U_{eff}$. One also needs a low vibronic coupling so that the effective barrier is closer to the total crystal field, rather than the first excited state. This hypothesis is consistent with

the interpretation of their own results in the record-setting dysprosocceniums with hysteresis up to 60 K[42] and 80 K[43]. Supporting this interpretation is also the fact that this correlation with $U_{eff,ff}$ is apparently absent in the case of $\tau_{QTM}$ (see Supplementary Fig. 36). Furthermore, this vibrationally controlled $U_{eff}$ would contribute to explain the typically weak correspondence between predicted (or even experimentally determined) energy levels and the $U_{eff}$ extracted from the spin relaxation experiments, a problem which is often minimised and sometimes justified by a role of QTM. Crucially, according to this idea, LnPc$_2$ and metallocenes would behave as exceptionally good nanomagnets not just because they provide exceptionally strong crystal field, but because they additionally provide exceptionally weak spin-vibrational coupling due to their rigidity, blocking Orbach and Raman processes simultaneously.

We have now obtained a likely rationalisation of why the controversial, oversimplified $U_{eff}$ is such a good predictor for the magnetic behaviour, and why a parameter that, resulting from a simplified fit, effectively summarises other relaxation mechanisms and correlates so unexpectedly well with the true $U_{eff,ff}$. The thermal dependence of the spin relaxation depends on Orbach + Raman, but $U_{eff}$ is statistically correlated with $C$ and, as can be speculated from the current understanding of spin relaxation[34], this can be related to the fact that $U_{eff}$ and $C$ are affected by the spin-vibrational relaxation. Whether the spin levels are real or virtual, to exchange energy with the thermal bath the spin needs to couple to vibrations. If this hypothesis is proven to be correct, a high $U_{eff}$ could be understood as acting as a witness for a weak spin-vibrational coupling.

## Influence of the coordination environment shape

Since we have established that $U_{eff}$ is a good predictor for magnetic behaviour and also rationalised how and why it correlates with Raman relaxation, let us now turn our attention to rational chemical design strategies. A key question is: are there coordination polyhedra that are intrinsically well suited to produce high effective barriers? Plotting all $U_{eff}$ values vs. the closest polyhedron for each complex, this seems to be the case. In contrast to what a cursory review of claims in literature would suggest, preparing lanthanide complexes that present a coordination environment close to D$_{4d}$ is not the best path. A more detailed analysis can be read in the Supporting Information Sections 7–8, including a critical assessment of data scarcity. Let us focus here on a salient case constituted by pentagonal bipyramids (with CN = 7), which present a striking distribution of $U_{eff}$, with consistently high values compared with any of the other common polyhedra (see Fig. 6a), as well as a high success rate both in terms of presenting a peak in χ″ at $10^3$ Hz and magnetic hysteresis (see Supplementary Fig. 32.2). Indeed, pentagonal bipyramids, much like square antiprisms, present no extradiagonal crystal-field terms therefore minimising spin mixing. Additionally, all of their diagonal terms are in first approximation protected from low-energy vibrations, minimising vibronic coupling (for a longer discussion of this see Supplementary Section 7). Their barriers can be maximised by vertical compression (see Supplementary Fig. 33).

The natural follow-up question would be how to chemically favour this kind of coordination rather than, for example, capped trigonal prisms, which also present CN = 7. From the dataset, it is obvious that employing non-chelating ligands massively favours the formation of pentagonal bipyramids (see Fig. 6b). The greatest synthetic competitor would seem to be octahedra, that also forms most often from ligands coordinating via a single atom, whereas most other shapes with CN ≥ 8 tend to result from chelating ligands. Similarly, a combination of oxygens and nitrogens is to be avoided, since for CN = 7 this tends to result in capped trigonal prisms; to obtain pentagonal bipyramids, an all-oxygen coordination sphere is often employed instead.

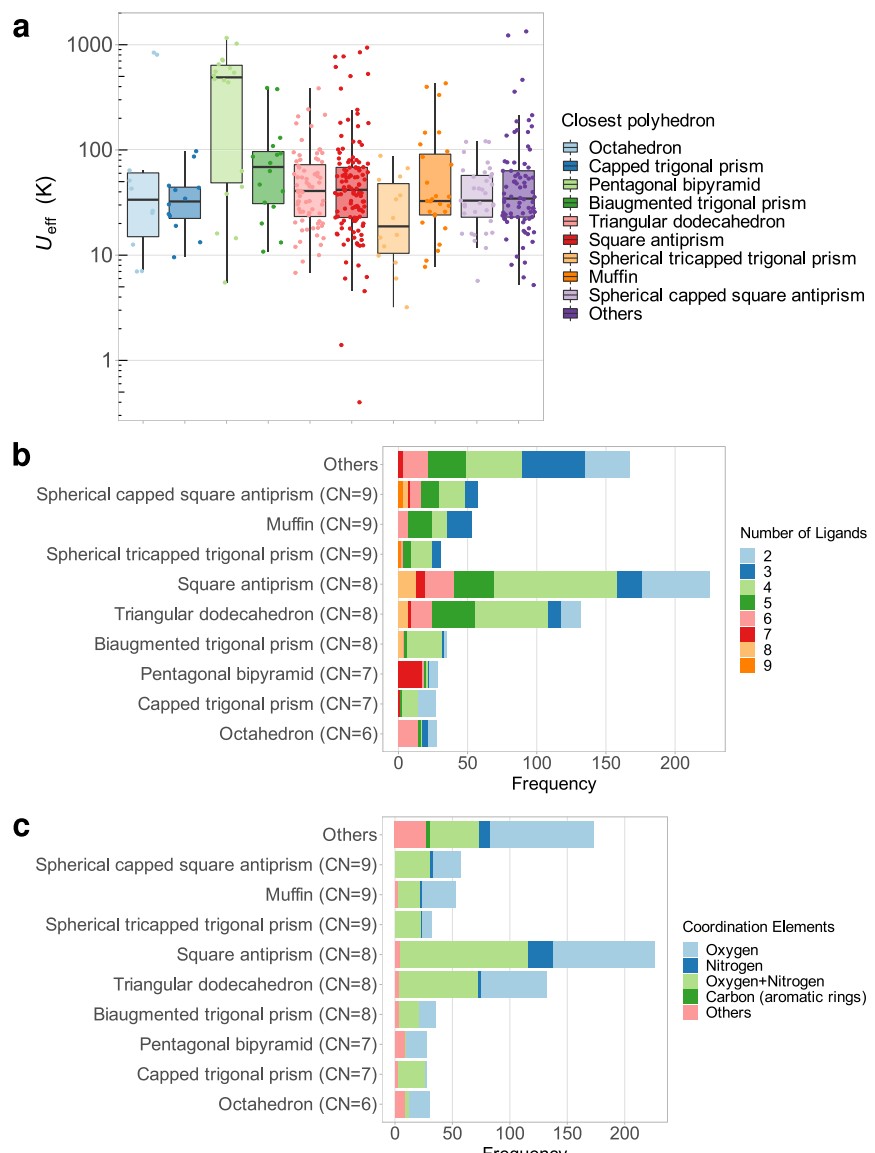

**Fig. 6 | Relation between $U_{\mathrm{eff}}$ and chemical design. a** Dependence between $U_{\mathrm{eff}}$ and coordination polyhedra. **b** Numbers of ligands that are present in the different coordination polyhedra. **c** Elements in the coordination sphere that are present in the different coordination polyhedra.

## Discussion

We have systematically analysed 448 articles to collect information from over 1400 samples reported over the first 17 years of the field of lanthanide-based SIMs and built a user-friendly dashboard for the visualisation of all the collected data. Moreover, we carried out an in-depth statistical analysis that allowed extracting trends, distinguishing the most relevant variables and grouping the data in clusters based on their chemical and physical properties. From this study, we can highlight two main pieces of information.

In the first place, from the point of view of the parametric characterisation, the simple Arrhenius fit assuming an Orbach process has been proven to be surprisingly meaningful, with the expected approximate relation between $\tau_0$ and $U_{\mathrm{eff}}$. One can therefore perform this oversimplified theoretical fit knowing that the effective energy barrier $U_{\mathrm{eff}}$ has been proven to present a consistently good correlation with SMM behaviour, as well as with Raman parameters $C, n$. Crucially, we have also shown the different nature of short-term magnetic memory in the form of the blocking temperature $T_{\mathrm{B3}}$ at $10^3$ Hz and its long-term counterpart in the form of maximum hysteresis

temperature $T_{\mathrm{hyst}}$. The best strategies that optimise the former are not necessarily the best for the latter.

In the second place, the chemical roadmap for the preparation of lanthanide complexes with higher $T_{\mathrm{hyst}}$ becomes now a little clearer. Generally, oblate ions are superior to prolate in terms of ac and $U_{\mathrm{eff}}$, but not in $T_{\mathrm{hyst}}$. So far, there has been a single chemical strategy to consistently and prolifically produce good magnetic memories, namely sandwiching an oblate ion between two rigid, planar, aromatic ligands; furthermore, the ion should be chosen to result in the most favourable $M_J$ structure, given the electron distribution offered by the ligand. Up to now, only two chemical families are well adapted to this strategy, namely TbPc$_2$ complexes and dysprosium metallocenes. Optimisation is ongoing within these two families, for example TbPc$_2$ complexes featuring a radical Pc display enhanced properties[44], and the reduced (divalent) analogues of DyCp*$_2$[45]. We find comparatively little value in further pursuing chemical strategies that have been amply explored and never yielded hysteresis above 10 K, like polyoxometalates, Schiff bases, diamagnetic transition

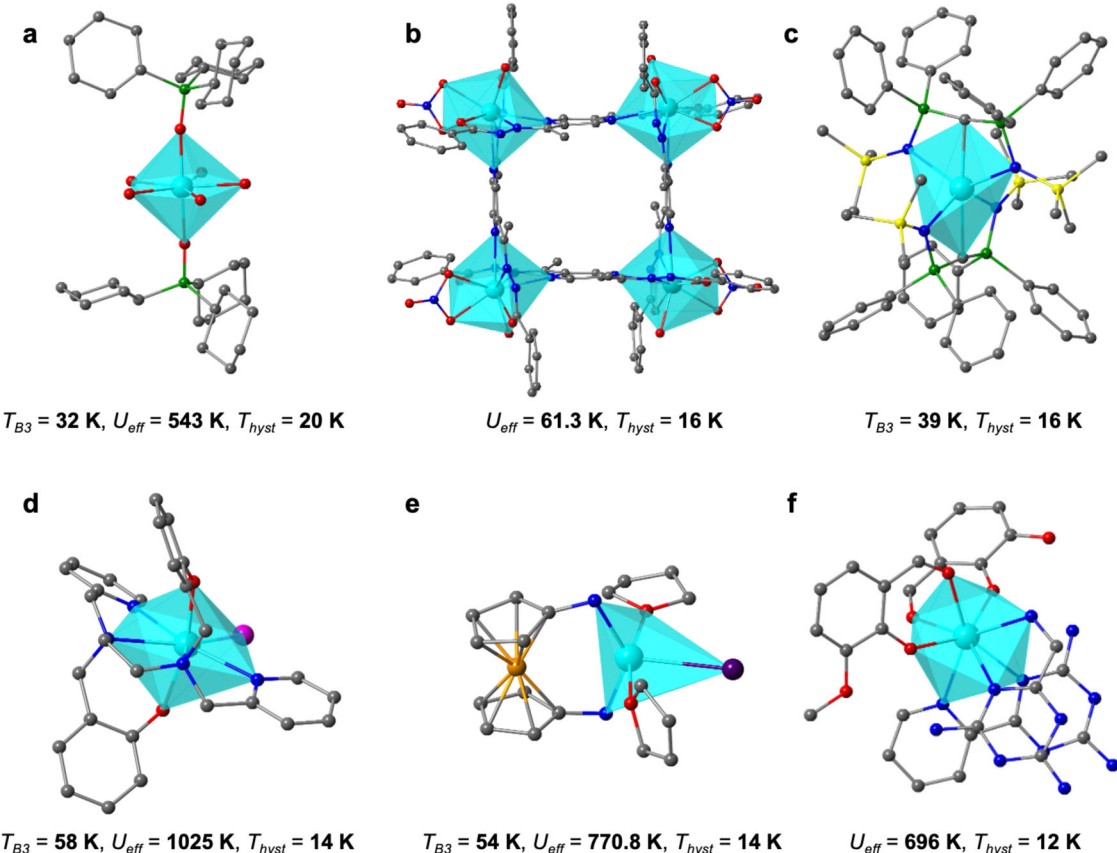

**Fig. 7 | Promising systems for the development of new high-$T_{hyst}$ SIMs, all chemically distinct from each other and from the TbPc$_2$ and metallocene categories**[47,74–78]. **a** [Dy(Cy$_3$PO)$_2$(H$_2$O)$_5$]$^{3+}$ (Cy$_3$PO = tricyclohexyl phosphine oxide)[47]; **b** [Dy$_4$(bzhdep-2H)$_4$(H$_2$O)$_4$(NO$_3$)$_4$] (bzhdep = pyrazine-2,5-diyl-bis(ethan−1-yl-1-ylidene)-di-(benzohydrazide))[74]; **c** [Dy(BIPM$^{TMS}$)$_2$]$^−$ (BIPM$^{TMS}$ = {C(PPh$_2$NSiMe$_3$)$_2$}$^{2−}$)[75]; **d** [Dy(bbpen)Br] (H$_2$bbpen = N,N′-bis(2-

hydroxybenzyl)-N,N′-bis(2-methylpyridyl)ethylenediamine)[76]; **e** (NN$^{TBS}$)DyI(THF)$_2$ (NN$^{TBS}$ = fc(NHSi$^t$BuMe$_2$)$_2$, fc = 1,1′-ferrocenediyl)[77]; **f** [DyLz$_2$(o-vanilin)$_2$]$^+$ (Lz = 6-pyridin-2-yl-[1,3,5]triazine-2,4-diamine)[78]. (colour scheme for atoms: green, P; cyan, Dy; grey, C; blue, N; yellow, Si; orange, Fe; red, O; magenta, Br; purple, I. Hydrogen atoms are not shown for clarity).

metals placed near the magnetic lanthanide, or radical ligands, except when acting as a bridge or as a part of a TbPc$_2$ complex. On the other hand, we also evidence that there is, of course, value in chemical ingenuity and exploration, in the quest for another successful strategy, which according to our results might well be based on equatorial erbium complexes, since these display consistently high $T_{hyst}$ values. Note that a few complexes included in our data fall into ample families such as mixed ligands or other families, and yet present excellent hysteresis temperatures. It is possible that the next family of record-setters is related to one of the promising candidates in Fig. 7. Two axial phosphine oxide ligands with bulky substituents seem to function in a similar way as metallocenes, despite the five equatorial H$_2$O molecules (see Fig. 7a)[46,47]. This strategy is not restricted to phosphine oxides and deserves to be explored further: complexes with 7 ligands have median values of $T_{hyst}$ close to 10 K, as high as those with 2 ligands. Indeed, and as pointed out above, axially compressed pentagonal bipyramids are a most promising yet underexplored strategy, and monodentate oxygen-based ligands seem to be a consistent path to achieve them.

At the same time, here we provide a catalogue of lanthanide SIMs, together with SIMDAVIS, a dashboard that allows its interactive navigation; this is a type of tool utterly missing in the field of molecular nanomagnets. Perhaps more importantly in the wider perspective of design of new materials[48–51] and new molecules[26,52], the dataset curated in this work will serve for machine-learning studies. It can also be employed as an annotated training dataset for the development of new

web scraping systems to retrieve chemical data[53,54], or even word embeddings[55], from the scientific literature. Finally, this work constitutes a step towards the availability of findable, accessible, interoperable, and reusable (FAIR) data in Chemistry[56].

## Methods
### Data gathering
This process started with the collection and organisation of literature data. The following search criterion was applied for the manuscript: articles are searched via Web of Science, employing this code:

TOPIC: TS = ((lanthan* OR 4\$f OR "rare\$earth") AND ((single NEAR/1 magnet*) OR "slow relaxation")) Timespan: 2003–2019.

For an article to be included in the study, it needs to contain data on at least one compound with certain requirements as follows: (a) contain one trivalent lanthanide ion from the set Ln = {Pr, Nd, Sm, Gd, Tb, Dy, Ho, Er, Tm, Yb} and (b) contain no other paramagnetic entity with the only accepted exception being the presence of a single radical in the coordination sphere and (c) present no strong Ln-Ln interaction, in particular meaning the Ln-Ln distance needs to be larger than 5 Å and more than 3 bridging atoms between neighbouring Ln centres, and there cannot be a radical in the bridge. Additionally, the data needs to include at least one of the following information: (a) whether $\chi''$ presents a maximum as a function of $T$, or a mere frequency-dependence, or neither; (b) $\chi''$ vs. $T$ with at least one frequency ($f$) in the window 0.9 kHz ≤ $f$ ≤ 1.1 kHz and at a field ($H$) in the window 0 ≤ $H$ ≤ 2 T; (c) $U_{eff}$; (d) the presence or absence of hysteresis; (e) $T_{hyst}$ at sweep speeds ($v$) below 0.3 T/s. The compounds were classified in chemical

families: LnPc$_2$, polyoxometalates, Schiff base, metallocenes, diketonates, radicals, TM near Ln, mixed ligands, and other families. Furthermore, we registered for each sample (when available), the lanthanide ion, its concentration, the coordination number and number of ligands coordinated to the lanthanide ions, the coordination elements, the presence of a field-dependent $\chi''$ or a maximum, the temperature of said maximum in presence or absence of an external magnetic field, the external magnetic field, the extracted effective energy barrier and relaxation time, either from a simplified Arrhenius fit or from a model considering all relaxation processes, whether these were extracted from the maxima of $\chi''$ vs. T at different frequencies or from an Argand fit, the presence of hysteresis in the magnetisation, and the maximum temperature at which it was recorded. Additionally the DOI, the full reference to the original article, and a link to a CIF file were recorded for each sample. The question of publication bias is addressed at the end of Supplementary Section 1.1. Further details including the classification in chemical families and the criteria for data extraction are provided in Supplementary Sections 1 and 2.

### SIMDAVIS dashboard
We programmed the dashboard using R language[57,58] and shiny[59], an open source R package to create the interactive web app. The design aimed to obtain a clean and simple user interface that adapts automatically to any screen size. The R packages readr[60], dplyr[61], DT[62], ggplot2[63] and rcrossref[64] were also employed in the development of the dataset or the app. The dashboard-style web application is available at https://go.uv.es/rosaleny/SIMDAVIS. This interface allows for variables in the analysis, and subsets of the data, to be adjusted and chosen in real time.

### Statistical analysis
The statistical analysis was also based on R, a widely used software environment for statistical computing and graphics, and included the Gifi system for Multiple Correspondence Analysis[65] (R homals package[66], ade4 package[67], see details in Supplementary Section 4.1), hierarchical clustering studies (FactoMineR[68], see details in Supplementary Section 4.2), lognormal modelling (Poisson's distribution, see Supplementary Section 4.3), factorial analysis of mixed data (FactoMineR and factoextra[69], see details in Supplementary Section 6), as well as Pearson's product-moment correlation and the Akaike information criterion (AIC)[70] (see details in Supplementary Section 5.3). The analysis was repeated and verified an overall excellent qualitative and quantitative consistency in all results between the period 2003-2017 (1044 samples) and 2003–2019 (1405 samples).

### SHAPE analysis
We employed a modified version of the pyCrystalField code[71] to extract the coordination environments of samples with a cif file in either the COD or the CCDC databases. We employed the SHAPE program to compare these with the reference polyhedra. Elongated and compressed versions of the reference polyhedra were also evaluated. We searched for correlations between the new data and the rest of the dataset.

## Data availability
The dataset collected and analysed during this study is freely available for download at https://go.uv.es/rosaleny/SIMDAVIS. Supplementary Data 1 file contains the SIMDavis dataset.

## Code availability
All custom code generated and employed for this study, namely the SIMDAVIS app version 1.1.9, is freely available for download at https://bitbucket.org/rosaleny/simdavis/src/master. In addition it can be downloaded as the Supplementary Software 1 (includes a SIMDAVIS Guide file).

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

## Acknowledgements

L.E.R., J.T.C. S.G.S., J.J.B., S.C.S and A.G.A. have been supported by the COST Action MolSpin on Molecular Spintronics (Project 15128). A.G.A. also thanks funding from European Union (EU) Programme Horizon 2020 (FATMOLS project) and Generalitat Valenciana (GVA) CIDEGENT/2021/018 grant. S.C.S acknowledges funding from the European Research Council (ERC) in EU Horizon 2020 grant agreement ERC-2017-AdG-788222 "MOL2D", and J.J.B. acknowledges grant agreement ERC-2021-StG-101042680 "2D-SMARTiES". S.C.S., S.G.S. and L.E.R. gratefully acknowledge support from the Spanish Ministerio de Ciencia e Innovación (MICINN) grant PID2020-117264GB-I00, and A.G.A. thanks MICINN PID2020-117177GB-I00 grant (both MICINN grants co-financed by FEDER funds). S.G.S. acknowledges MICINN PRE2018-083350 grant related to MINECO CTQ2017-89528-P project. S.C.S. acknowledges Excellence Unit María de Maeztu CEX2019-000919-M funding. J.T.C. thanks funding to the Fundação para a Ciência e a Tecnologia (projects UIDB/04044/2020 and UIDP/04044/2020). L.E.R. gratefully acknowledges support from GVA PROMETEO/2019/066, J.J.B. thanks GVA CDEIGENT/2019/022 grant. A.S. acknowledges that this research used resources at the Spallation Neutron Source, a DOE Office of Science User Facility operated by the Oak Ridge National Laboratory. The statistical analysis was performed by Raquel Gavidia Josa with the Statistical Section of the S.C.S.I.E. (Universitat de València).

## Author contributions

All authors contributed to the different stages of the work plan as detailed below. A.G.A. suggested the starting point of the analysis, with contributions from J.J.B. and S.C.S. J.T.C. and Y.D. designed the whole procedure for raw data extraction and classification. Y.D., J.T.C., A.G.A., S.C.S, S.G.S. and J.J.B. did the manual data-mining. Y.D., J.T.C., A.G.A., L.E.R. and S.C.S. double-checked the raw data. A.S. adapted pyCrystalField and extracted the coordination environments. A.S. and S.G.S. performed all calculations. L.E.R. and A.G.A. cleaned and organised the raw data into a tidy dataset. L.E.R. and A.G.A. conceived and A.G.A. supervised the statistical data analysis. L.E.R. conceived and programmed the dashboard-style interactive web application for data visualisation and analysis. All authors contributed to the preparation of the manuscript.

## Competing interests

The authors declare no competing interests.
