## [Peer Review File · Nature Communications]

Data-driven design of molecular nanomagnetsEditorial Note: This manuscript has been previously reviewed at another journal that is not operating a transparent peer review scheme. This document only contains reviewer comments and rebuttal letters for versions considered at *Nature Communications*.

REVIEWER COMMENTS

Reviewer #1 (Remarks to the Author):

The manuscript has now been strongly revised by the authors and contains a new discussion of possible chemical strategies in view of improved SIMs based on the statistical analysis of literature.

As I expressed in my previous report, this is an important study for several reasons. From a methodological point of view the use of data mining and literature screening to this extent is a complete novelty for the field of molecular magnetism. Moreover, the authors have now improved the qualitative analysis of literature and provide a more convincing discussion of how the field can exploit the results of the study.

I only have one minor comment for the authors. The new manuscript now heavily make use of terms such as vibronic coupling, ligand rigidity, etc. As such I feel that several of the key publications on this topic from the last few years are missing a reference. The authors might want to consider improving the bibliography.

It is reasonable to expect that this work will have a good impact on the community and as such I recommend it for publication in *Nature Communications*.

Reviewer #2 (Remarks to the Author):

The paper describes a huge effort of compiling data stemming from more than 1400 lanthanide-based nanomagnets that were published in articles within 2003 and 2019. This extracted data served to devise a dashboard-style web application called SIMDAVIS. Impressively, the manuscript states that the dataset contains over 10000 and 5000 independent pieces of chemical and physical information, respectively.

The manuscript has substantially improved compared to the previous version. SIMDAVIS will be a powerful tool indeed that the community will use. Rephrasing the manuscript also helped significantly not to make the paper sound like a manual for a program. It is nice to see that the Arrhenius barrier correlates well with magnetic memory since both Orbach and Raman processes can be controlled by vibronic coupling.

The study clearly shows that the program provides most accurate results for bis-phthalocyaninato sandwiches and metallocenes. However, it should be distinguished in the paper between mono- and multinuclear metallocenes. Studied are here only metallocenes with one metal ion. Both endohedral metallofullerenes and radical SMMs are not considered, but both can contain metallocenes too and a fullerene does consist of metallocene fragments. Since SIMDAVIS considers primarily single-ion magnets it would be best if the title of the manuscript already reflects that, instead of just saying "molecular nanomagnets". The fact that mostly single-ion magnets are used, should also be mentioned in the abstract and introduction of the manuscript.

Corrections to make:

Page 2: change "from a strong spin-orbit coupling" to "from strong spin-orbit coupling"

Fig. 1 a: Although mentioned further down in the manuscript, it would be beneficial to include the definition of TB3H into the figure caption of Fig. 1a.

Fig. 1 c caption: change "(purple), signalling a fast relaxation at zero magnetic field" to "(purple), where latter signals a fast relaxation at zero magnetic field"

Fig. 2 b: Why is the coordination number here 2? The metal is interacting with 10 carbon atoms. Typically, a Cp ring is considered to occupy three coordination sites of the lanthanide. So, coord. number should be six. Or if all five carbon atoms are considered as interacting with the metal, then coord. number should be 10. This needs to be addressed in the manuscript as otherwise the used terminology here will be considered as wrong from an organometallic chemistry perspective.

Page 6, line 148: change "are supposed to be correlated" to "are supposed to correlate"

Page 7, line 194: do not just use DyCp₂ as that is technically wrong since just Cp implies the bare cyclopentadienyl ligand, however, the Cp systems used in this study bear substituents

Page 7, line 210: change "This means there is" to "This means that there is"

Taken together, the manuscript is a nice, very valuable study that after fixing my relatively minor comments above, I recommend publication in Nature Communications.

Reviewer #3 (Remarks to the Author):

The revised version of Duan et al represents an improved version of the original manuscript, having successfully addressed most, but not all, of my previous concerns. Overall, I appreciate the efforts made by the authors to provide an even better discussion and application. However, I am not convinced by one of the main new additions, namely the alleged correlation found between the Orbach and Raman parameters or the explanation provided, as I consider that such claims need a more statistically firm analysis. As such, I recommend publication after major revision. Please see my comments below.

Major:

The main concern of the revised version relates to the new Raman part. Particularity

- Figure 5 summarises most of my concerns:

- What metric do the authors use to assess that $U_{\text{eff,ff}}$ and C correlate? Looking at Figure 5, said correlation is not obvious at all, as there is a huge dispersion. For instance, the four data points related to "Carbon (aromatic rings)" category with the largest $U_{\text{eff,ff}}$ span ca six orders of magnitude in the C values. To my account, these two parameters are not correlated, and the authors should provide more than just a graphic representation to back such a strong statement.
- Following on the previous comment, one wonders why the authors did not include C and n in the study of "5.3. Correlations between physical variables" to provide a more sound statistical analysis?
- Would it be useful to calculate the area spanned by the confidence intervals to make the comparison between different correlations easier?
- Why are the confidence intervals shown in most of the Supplementary Figures of Section 8.2, but not in Figure 5?
- The authors only present in the main text the correlation between U_{eff} and C. Why would U_{eff} not correlate with n, or τ_0 with C? I'd argue that Supplementary Figure 34c shows that $U_{\text{eff,ff}}$ correlates better with n than with C actually. Again, having a metric that allows objective comparison would improve this part of the study.
- Again on comparing different correlations, the authors say that "Supporting this interpretation is also the fact that this correlation with $U_{\text{eff,ff}}$ is apparently absent in the case of τ_{QTM} ". In all honesty, to me, this looks like a better correlation than that of Figure 5.
- As such, the authors need to still present compelling evidence to be able to back claims like " U_{eff} is correlated with C", "Since we have [...] rationalised how and why (U_{eff}) correlates with Raman relaxation" or " U_{eff} has been proven to present a consistently good correlation with [...] Raman parameters C, n."

- When trying to reproduce Fig5 in the dashboard, it shows far fewer data points. Why is this?

- When providing an explanation to “But these are supposed to be two fully independent mechanisms, so what could be the reason behind this apparent coincidence?” the authors conclude that the correlation is due to spin-phonon coupling. I have two issues with this: first, the fact that these might actually not be correlated (see above) and second, concerning the explanation provided for the alleged correlation, i.e. spin-phonon coupling. This explanation, together with the sentences “Thus a high U_{eff} can be understood as acting as a witness for a weak spin-vibrational coupling” in the main text and “This supports the hypothesis we expose in the main text of high values of U_{eff} being a witness for weak vibronic couplings, which also translate into low values of C, n ” in the supplementary, implies that the magnitudes of spin-phonon coupling in the Orbach regime (optical phonons) and Raman regime (acoustic phonons) are correlated. This is an unfounded assumption, and no justification is provided in the text. As such, I suggest that the authors i) perform a better statistical analysis of the correlation between the Raman and Orbach parameters, and ii) if they do correlate, simply present the finding not trying to rationalise it in vague terms of spin-phonon coupling.

- Following on criticism raised in the previous version, the authors should make an effort at introducing key concepts. The sentence “Initially, oblate ions Tb and Dy [...] prolate ions, e.g. Er, Tm, Yb” again catches the reader by surprise, and key references to this electrostatic design criteria are missing. It does not help that these terms are often used throughout the text. Also, talking about oblate or prolate ions is confusing, as it all depends on the environment the ion is encapsulated in. The ions are neither oblate nor prolate, it is the electron density associated with different m_j projections the one showing an angular dependence. If you put a Dy^{3+} ion in a rhombic environment, the least magnetic prolate-like state will be the ground state. This obviously affects the “ L_n anisotropy” variable.

- The authors argue that violin plots are much more informative than box plots, especially when the data presents more than one maximum. Still, in the app, there is no “Violin plot” tab. I suggest the app would be an even better tool if this option is implemented.

Minor

- I'd be wary of the sentence “The consideration of other processes in the fit, such as the Raman process and Quantum Tunneling of the Magnetisation (QTM), in principle results in more accurate values”, which can be interpreted as a recommendation to always use more fitting parameters. Given that the literature is already filled with examples of over parametrisation, a clarification that this approach is to be taken only if one has enough data points in each of the regimes would be appreciated.

- I still find the claim “The best metric for slow relaxation is hysteresis temperature” problematic. As discussed, this depends on the sweeping rate, which is by no means standardised. I suggest that it would be less controversial to convey the message that open hysteresis is a necessary condition for SMMs properties.

- It is surprising that the authors do not reference Neese's work (Faraday Discuss., 2011, 148, 229-238) when discussing “to improve the maximum χ_{Thy} is to maximise the total effective spin (S), rather than the magnetic anisotropy (D).”, which demonstrates that S and D are not independent.

- Fig 1a suggests that applying an external magnetic field should always be done as it leads to a maximum in the ac. I presume the authors just wanted to represent that in-field measurements remove QTM, but this is not at all obvious from reading the caption. Also, I would argue that a system that only shows slow relaxation of magnetisation in-field is not really a SMM. I suggest this figure or its explanation to be improved. In fig1b, the different elements of the fitting function should be aligned. Fig 1d is not mentioned in the main text, so one wonders what purpose it serves.

- I'd argue that the concept “ac magnetometry” is incorrect and suggest to use “ac susceptometry”

instead.

- The sentences “according to a simplified two-phonon Orbach model” and “An approximate relation between τ_0 and U_{eff} can be derived for two-phonon Orbach process within a Debye phonon model” need to be referenced.

- With respect to section 5.5 in the supplementary material, the derivation is much clearer now. As stated by the authors, Eq. 1 models a two-phonon Orbach process with a Debye model for the description of the phonons. Equating this to the generally employed Eq. 2 implies collapsing all the multiple-phonon processes encapsulated in U_{eff} to two-phonon processes. I suggest the authors comment on this and on the possibility that deviations from an exponent of 3 in Eq 5 might arise from the fact that what governs the overall reversal of magnetisation is not entirely due to two-phonon processes. Also, more of a question than a comment, does equating these two equations imply that the relaxation path occurs through the 2nd excited Kramers doublet?

- It's odd to find copy pasted comments made by the referees in the main text: “historically, the fit to the Orbach process has been applied to the relaxation times obtained at the highest temperatures, even in systems where at very low temperatures the relaxation times were indicating purely quantum tunnelling or Raman relaxation mechanism.”.

- In the violin plots, why are the upper and lower adjacent values not shown? Is the median represented by the middle horizontal line?

- The two statements in the sentence “Our analysis showed that the Arrhenius energy barrier correlates unexpectedly well with the magnetic memory, as both Orbach and Raman processes can be controlled by vibronic coupling.” feel disconnected, as there is no clear logical link between them.

-One of the exciting prospects of having tools like this is to update them with new information to validate or challenge previous assumptions and conclusions. In that sense, I have the impression that this data set would benefit enormously from future phonon calculations, at least on the best-performing SMMs, especially to back claims made about spin-phonon coupling. Is this something the authors would consider to discuss?

- I'd suggest that when selecting specific categories to represent in the app, only those appear labelled in the plot. For instance, in Supplementary Figure 11, only the D_y and T_b labels should appear.

- I find the question “to determine whether one would need to employ U_{eff} instead.” ill-defined, as it is often the case that the relaxation profile only contains data on the Orbach mechanism, so accounting for the Raman and QTM parameters that would yield U_{eff} is just not possible. Maybe the authors can clarify this.

- The sentence “the case of τ_{QTM} (see Supplementary Figure 35).” Should read as “the case of τ_{QTM} (see Supplementary Figure 36).”

REVIEWER COMMENTS

Reviewer #1 (Remarks to the Author):

The manuscript has now been strongly revised by the authors and contains a new discussion of possible chemical strategies in view of improved SIMs based on the statistical analysis of literature.

As I expressed in my previous report, this is an important study for several reasons. From a methodological point of view the use of data mining and literature screening to this extent is a complete novelty for the field of molecular magnetism. Moreover, the authors have now improved the qualitative analysis of literature and provide a more convincing discussion of how the field can exploit the results of the study.

We thank reviewer 1 for his comments on the study's quality and novelty.

I only have one minor comment for the authors. The new manuscript now heavily make use of terms such as vibronic coupling, ligand rigidity, etc. As such I feel that several of the key publications on this topic from the last few years are missing a reference. The authors might want to consider improving the bibliography.

According to reviewer suggestions we have included a few recent key references in the text, namely:

Lunghi *et al.* Intra-molecular origin of the spin-phonon coupling in slow-relaxing molecular magnets, *Chem. Sci.*, 2017, **8**, 6051-6059
<https://doi.org/10.1039/c7sc02832f>

A. Ullah *et al.* In Silico Molecular Engineering of Dysprosocenium-Based Complexes to Decouple Spin Energy Levels from Molecular Vibrations, *J. Phys. Chem. Lett.*, 2019, **10**, 7678–7683
<https://doi.org/10.1021/acs.jpcllett.9b02982>

D.H. Moseley *et al.* Inter-Kramers Transitions and Spin–Phonon Couplings in a Lanthanide-Based Single-Molecule Magnet, 2020, *Inorg. Chem.*, **59**, 5218-5230
<https://doi.org/10.1021/acs.inorgchem.0c00523>

A. Castro-Alvarez *et al.* High performance single-molecule magnets, Orbach or Raman relaxation suppression? *Inorg. Chem. Frontiers*, 2020, **7**, 2478-2486
<https://doi.org/10.1039/D0QI00487A>

D. Aravena and E. Ruiz. Spin dynamics in single-molecule magnets and molecular qubits. *Dalton Trans.*, 2020, **49**, 9916-9928
<https://doi.org/10.1039/D0DT01414A>

J.G.C. Kragoskow *et al.* Analysis of vibronic coupling in a 4f molecular magnet with FIRMS. Nat. Comms., 2022, **13**, 825
<https://doi.org/10.1038/s41467-022-28352-2>

We now state in the Results section “Orbach mechanism: oversimplified, predictive... a function of vibronic coupling?” (page 11):

“However, there is also a possible physical reason behind the correlation we found. It is the coupling between spin states and vibrations: a recently recognized key parameter in molecular spin dynamics.^{39–41} Spin-vibration coupling is a common factor for both relaxation pathways, and plays a vital role in both Raman^{41,42} and Orbach^{41,43,44} mechanisms.”

It is reasonable to expect that this work will have a good impact on the community and as such I recommend it for publication in Nature Communications.

Reviewer #2 (Remarks to the Author):

The paper describes a huge effort of compiling data stemming from more than 1400 lanthanide-based nanomagnets that were published in articles within 2003 and 2019. This extracted data served to devise a dashboard-style web application called SIMDAVIS. Impressively, the manuscript states that the dataset contains over 10000 and 5000 independent pieces of chemical and physical information, respectively.

The manuscript has substantially improved compared to the previous version. SIMDAVIS will be a powerful tool indeed that the community will use. Rephrasing the manuscript also helped significantly not to make the paper sound like a manual for a program. It is nice to see that the Arrhenius barrier correlates well with magnetic memory since both Orbach and Raman processes can be controlled by vibronic coupling.

We thank reviewer 2 for considering our several improvements in the new version of the manuscript.

The study clearly shows that the program provides most accurate results for bis-phthalocyaninato sandwiches and metallocenes. However, it should be distinguished in the paper between mono- and multinuclear metallocenes. Studied are here only metallocenes with one metal ion. Both endohedral metallofullerenes and radical SMMs are not considered, but both can contain metallocenes too and a fullerene does consist of metallocene fragments. Since SIMDAVIS considers primarily single-ion magnets it would be best if the title of the manuscript already reflects that, instead of just saying “molecular nanomagnets”. The fact that mostly single-ion magnets are used, should also be mentioned in the abstract and introduction of the manuscript.

Corrections to make:

Page 2: change “from a strong spin-orbit coupling” to “from strong spin-orbit coupling”

Done. This sentence now reads: “from strong spin-orbit coupling which”

Fig. 1 a: Although mentioned further down in the manuscript, it would be beneficial to include the definition of TB3H into the figure caption of Fig. 1a.

We thank the referee for this clarification, we have reformulated the text, now it reads: “being T_{B3H} the temperature for the maximum χ ” at 10^3 Hz in the presence of a magnetic field.”

Fig. 1 c caption: change “(purple), signalling a fast relaxation at zero magnetic field” to “(purple), where latter signals a fast relaxation at zero magnetic field”

Done. Now the sentence reads:

“Magnetic hysteresis can be full (orange) or “pinched”, also known as “butterfly” (purple), where the latter signals a fast relaxation at zero magnetic field”

Fig. 2 b: Why is the coordination number here 2? The metal is interacting with 10 carbon atoms. Typically, a Cp ring is considered to occupy three coordination sites of the lanthanide. So, coord. number should be six. Or if all five carbon atoms are considered as interacting with the metal, then coord. number should be 10. This needs to be addressed in the manuscript as otherwise the used terminology here will be considered as wrong from an organometallic chemistry perspective.

We thank the referee for raising this question, that indeed requires clarification. We added the following explanation to the Supplementary Section 1:

“From the point of view of steric hindrance in coordination chemistry, a Cp ring can be considered as equivalent to 3 coordination sites. However, from the point of view of the Crystal Field it creates and its coupling to “soft” vibrations, it is well-know that the electronic density responsible for the coordination is the π -electron cloud. Moreover, this π -electron cloud has a very reduced freedom to distort, due to the rigid nature of the aromatic ring. This is why we chose to treat the whole orbital (which in fact is delocalized over the rigid ligand) as a single coordination site for the purposes of the present study.”

We refer to this section in the caption of Figure 2b:

“**b**, Metallocene complex LnCp^*_2 ($T_{\text{hyst}} = 60 \text{ K}$),²³ see Supplementary Information 1 for our CN criterion for the metallocene family.”

Page 6, line 148: change “are supposed to be correlated” to “are supposed to correlate”

Done. Now the sentence reads

“according to a simplified two-phonon Orbach model, the two variables are supposed to correlate.”

Page 7, line 194: do not just use DyCp₂ as that is technically wrong since just Cp implies the bare cyclopentadienyl ligand, however, the Cp systems used in this study bear substituents

We thank the referee for pointing out this nomenclature error. According to the abbreviation by the original authors we have modified the simplified name to “DyCp*₂”.

Page 7, line 210: change “This means there is” to “This means that there is”

Done. Now the sentence read as follows:

“This means that there is a limit on the information one can independently extract from the rest of the chemical variables.”

Taken together, the manuscript is a nice, very valuable study that after fixing my relatively minor comments above, I recommend publication in Nature Communications.

We again thank the reviewer for his suggestions, proposed fixes and comments.

Reviewer #3 (Remarks to the Author):

The revised version of Duan et al represents an improved version of the original manuscript, having successfully addressed most, but not all, of my previous concerns. Overall, I appreciate the efforts made by the authors to provide an even better discussion and application. However, I am not convinced by one of the main new additions, namely the alleged correlation found between the Orbach and Raman parameters or the explanation provided, as I consider that such claims need a more statistically firm analysis. As such, I recommend publication after major revision. Please see my comments below.

We thank the reviewer for their overall assessment but also for the criticism which has allowed us to revisit our conclusion with a serious statistical analysis, as it deserves. This new analysis has been added in the Supplementary Information Sections 9.3-9.5: ~12 more SI pages and 13 more SI figures. This new analysis can also be found on the annex on this Answer to Referees document (pages 14-28). See more detailed explanation below.

Major:

The main concern of the revised version relates to the new Raman part. Particularity - Figure 5 summarises most of my concerns:

- What metric do the authors use to assess that $U_{\text{eff,ff}}$ and C correlate? Looking at Figure 5, said correlation is not obvious at all, as there is a huge dispersion. For instance, the four data points related to “Carbon (aromatic rings)” category with the largest $U_{\text{eff,ff}}$ span ca six orders of magnitude in the C values. To my account, these two parameters are not correlated, and the authors should provide more than just a graphic representation to back such a strong statement.

The new Supplementary Information Section 9.3 is devoted to this, initially quantifying the linear correlation between $\log(U_{\text{eff,ff}})$ and $\log(C)$ but also applying several independent statistical tests to confirm the validity of assuming this linear correlation, as detailed below. We can see that the behaviour of the variables is very similar to a normal distribution, and that the calculated correlation coefficient is significant with a value of -0.729, which indicates that the association between the variables is high. Pearson's test resulted in a $p \sim 10^{-8}$ for the hypothesis of 0 correlation. The linear model also obtained $p < 0.001$.

We verified all standard conditions of applicability of this linear model, and in all cases the result was positive. In detail: we checked the linearity of the model (residuals vs fitted values) and found that the residuals are randomly distributed around 0 so linearity is accepted. We checked the normality of the model (standardised residuals vs their theoretical values) and found that for the Shapiro-Wilk normality test the p-value obtained (0.67) is not significant, so we are not in a position to reject the normality hypothesis, thus the normal distribution of the residuals is confirmed. We checked the homogeneity of variance (square root of the standardised residuals vs the fitted values), where the Breusch-Pagan test resulted in a non-significant value (0.06), meaning we accept the condition of homogeneity of variances. Finally, we checked the autocorrelation of residuals (standardised residuals vs leverage) which also do not show any trend.

As a minor but still important point, since Figure 5, and many others, are in log-log form, they actually reflect the correlation between the logarithms of the variables, not the variables themselves, hence the many orders of magnitude of variation rightfully pointed out by the referee. In particular, the linear model found that a one-unit increase in the predictor $\log(U_{\text{eff}})$ causes an average change of -3.038 units in the response variable $\log(C)$. The manuscript and the SI have been corrected in that regard, and now either states “in logarithmic form” (Results Section “Orbach mechanism: oversimplified, predictive... a function of vibronic coupling?”, pages 10-11) or explicitly discusses the logarithms of the variables rather than the variables themselves, e.g. we state:

“We found that $\log(U_{\text{eff}})$ seems to correlate quite well with $\log(C)$ (see Supplementary Figure 34, top), and this correlation is perhaps more clear with $\log(U_{\text{eff,ff}})$ (see Figure 5).”

• Following on the previous comment, one wonders why the authors did not include C and n in the study of “5.3. Correlations between physical variables” to provide a more sound statistical analysis?

This is a good point, and it is now addressed, see Supplementary Information Sections 9.3 and 9.4 (available as Annex in the present answer to the referees document). We however distinguished between the variables discussed in Supplementary Section 5, i.e. within the Orbach modelling, and the inter-modelling analysis in section 9.

• Would it be useful to calculate the area spanned by the confidence intervals to make the comparison between different correlations easier?

We stuck to other procedures to evaluate the validity of the correlations see Supplementary Information Sections 9.3-9.5.

• Why are the confidence intervals shown in most of the Supplementary Figures of Section 8.2, but not in Figure 5?

Figure 5 was redone (without the unnecessary categorization), and now includes the confidence interval:

Figure 5 | Relation between $U_{\text{eff,ff}}$ and Raman relaxation parameter C , in logarithmic form. Data points are accompanied by a least square fit including confidence intervals.

- The authors only present in the main text the correlation between U_{eff} and C . Why would U_{eff} not correlate with n , or τ_{0} with C ? I'd argue that Supplementary Figure 34c shows that $U_{eff,ff}$ correlates better with n than with C actually. Again, having a metric that allows objective comparison would improve this part of the study.

Having established that $U_{eff,ff}$ is a better predictor than τ_{0} for the overall magnetic behaviour, we systematically checked the correlations between $\log(U_{eff,ff})$ and the logarithms of parameters characterising other relaxation mechanisms: $\log(C)$, $\log(n)$ and $\log(\tau_{QTM})$, see Supplementary Information Sections 9.3-9.5.

The statistical tests we applied did find a correlation between $\log(U_{eff,ff})$ and $\log(n)$ and a correlation was found, but weaker than in the case of $\log(C)$, (e.g. the regression changes by removing the outliers for $\log(U_{eff,ff})$ vs $\log(n)$, while in the case of $\log(U_{eff,ff})$ vs $\log(n)$ the regression is robust); see new Supplementary Section 9.4 or the Annex below.

- Again on comparing different correlations, the authors say that “Supporting this interpretation is also the fact that this correlation with $U_{eff,ff}$ is apparently absent in the case of τ_{QTM} ”. In all honesty, to me, this looks like a better correlation than that of Figure 5.

See above. The statistical tests we applied did not find a correlation between $\log(U_{eff,ff})$ and $\log(\tau_{QTM})$, see new Supplementary Section 9.5 or the Annex below.

- As such, the authors need to still present compelling evidence to be able to back claims like “ U_{eff} is correlated with C ”, “Since we have [...] rationalised how and why (U_{eff}) correlates with Raman relaxation” or “ U_{eff} has been proven to present a consistently good correlation with [...] Raman parameters C , n ”.

Indeed, the previous version of the manuscript did not include compelling evidence, but our current statistical analysis demonstrates this, see above (and new Supplementary Information Sections 9.3-9.5 or the Annex below).

- When trying to reproduce Fig5 in the dashboard, it shows far fewer data points. Why is this?

We are not able to reproduce the discrepancy. We include here for visual comparison the previous version of Fig 5 (right panel figure above) where $U_{\text{eff,ff}}$ is represented as a function of C , and the data points are colored by Coordination Elements, and the same plot produced by the dashboard at the time of this writing (left panel figure above). When we have run into similar problems, it has sometimes been due to selecting a different “Color by” variable, e.g. if “closest polyhedron” has been chosen as a “Color by” variable, only 21 samples present information on $U_{\text{eff,ff}}$, C and Closest polyhedron, whereas 48 samples present information on $U_{\text{eff,ff}}$, C and Coordination elements. Plotting U_{eff} (rather than $U_{\text{eff,ff}}$) vs C also results in a mere 19 samples, but again this is only speculation.

- When providing an explanation to “But these are supposed to be two fully independent mechanisms, so what could be the reason behind this apparent coincidence?” the authors conclude that the correlation is due to spin-phonon coupling. I have two issues with this: first, the fact that these might actually not be correlated (see above) and second, concerning the explanation provided for the alleged correlation, i.e. spin-phonon coupling. This explanation, together with the sentences “Thus a high U_{eff} can be understood as acting as a witness for a weak spin-vibrational coupling” in the main text and “This supports the hypothesis we expose in the main text of high values of U_{eff} being a witness for weak vibronic couplings, which also translate into low values of C , n ” in the supplementary, implies that the magnitudes of spin-phonon coupling in the Orbach regime (optical phonons) and Raman regime (acoustic phonons) are correlated. This is an unfounded assumption, and no justification is provided in the text. As such, I suggest that the authors i) perform a better statistical analysis of the correlation between the Raman and Orbach parameters, and ii) if they do correlate, simply present the finding not trying to rationalise it in vague terms of spin-phonon coupling.

It is true that in the previous version of the manuscript we provided no statistical analysis for this apparent correlation, now we do (Supplementary Information Sections 9.3-9.4). It is also true that the rationalisation is for the moment speculative, however note that it is not theoretically unexpected in the field. On the contrary, it can be said that this is in line with the current understanding of these processes; note the comment of reviewer #2: “It is nice to see that the Arrhenius barrier correlates well with magnetic memory since both Orbach and Raman processes can be controlled by vibronic coupling”. Following the advice of reviewer #1, we now include multiple references to evidence the current consensus (pages 2-3 on this Answer to Referees document). We also are now more clear in the fact that while the statistical correlation is well established, the vibronic hypothesis would need to be verified, and in the Results Section “Orbach mechanism: oversimplified, predictive... a function of vibronic coupling?”, pages 11-12, we now state:

“We have now obtained a likely rationalisation of why the controversial, oversimplified U_{eff} is such a good predictor for the magnetic behaviour, and why a parameter that, resulting from a simplified fit, effectively summarises other relaxation mechanisms and correlates so unexpectedly well with the true $U_{\text{eff,ff}}$. The thermal dependence of the spin relaxation depends on Orbach+Raman, but U_{eff} is statistically correlated with C and, as can be speculated from the current understanding of spin relaxation,³⁸ this can be related to the fact that U_{eff} , C are heavily controlled by the spin-vibrational relaxation. Whether the spin levels are real or virtual, to exchange energy with the thermal bath the spin needs to couple to

vibrations. If this hypothesis is proven to be correct, a high U_{eff} could be understood as acting as a witness for a weak spin-vibrational coupling.”

Similarly, in the abstract we now state:

Our analysis showed that the Arrhenius energy barrier correlates unexpectedly well with the magnetic memory. Furthermore, as both Orbach and Raman processes can be controlled by vibronic coupling, chemical design of the coordination scheme may be used to reduce the relaxation rates. Indeed, only bis-phthalocyaninato sandwiches and metallocenes, with rigid ligands, consistently present magnetic memory up to high temperature.

- Following on criticism raised in the previous version, the authors should make an effort at introducing key concepts. The sentence “Initially, oblate ions Tb and Dy [...] prolate ions, e.g. Er, Tm, Yb” again catches the reader by surprise, and key references to this electrostatic design criteria are missing. It does not help that these terms are often used throughout the text. Also, talking about oblate or prolate ions is confusing, as it all depends on the environment the ion is encapsulated in. The ions are neither oblate nor prolate, it is the electron density associated with different m_j projections the one showing an angular dependence. If you put a Dy^{3+} ion in a rhombic environment, the least magnetic prolate-like state will be the ground state. This obviously affects the “Ln anisotropy” variable.

Herein we follow Long’s criterion [Rinehart, Long, *Chem. Sci.*, 2011, **2**, 2078-2085] of assigning “oblate” and “prolate” characters to lanthanide ions, which corresponds exactly with Stevens α parameter. In the simplified zero field splitting Hamiltonian, this means that for an oblate ion (such as Dy^{3+} or Tb^{3+}) an axial coordination favours a high M_j ground state, whereas for a prolate ion (notably, Er^{3+}) it is the equatorial coordination the one that favours a high M_j ground state. This notation is useful for our dataset, since it opens the door to future studies relating this oblate/prolate character with our characterisation of the axial distortion.

The Supplementary Information section 1 “Construction of the dataset” (page 5), where this variable is explained now reads:

“-The parameter “Ln anisotropy” is categorical and takes one of the following 3 values for each sample: {prolate [0]; oblate [1]; isotropic [2]}. This is determined directly by the Ln ion and presents a 1-to-1 correspondence with the sign of the Stevens α parameter. In the simplified zero field splitting Hamiltonian, this means that for an oblate ion (such as Dy^{3+} or Tb^{3+}) an axial coordination favours a high m_j ground state, whereas for a prolate ion (notably, Er^{3+}) it is the equatorial coordination the one that favours a high M_j ground state.”

Additionally, in the first mention of the oblate/prolate terminology in the main text (Section “A brief history of SIMs”, pages 3-4), we indicate:

“Initially, Tb^{3+} and Dy^{3+} ions were the most commonly studied. These present an equatorially expanded f -electron charge cloud and are known as “oblate” (see Supplementary Section 1). Success cases were also found for lanthanide ions with axially elongated f -electron charge cloud (prolate ions, e.g. Er^{3+} , Tm^{3+} , Yb^{3+}).”

- The authors argue that violin plots are much more informative than box plots, especially when the data presents more than one maximum. Still, in the app, there is no "Violin plot" tab. I suggest the app would be an even better tool if this option is implemented.

We agree with the referee: violin plots may show more information than regular box plots, and this can be useful when the data points are multimodally distributed. However, as reviewer #2 from the previous refereeing round stated, violin plots can be very difficult to interpret for some inexperienced readers. In this specific case, we think that violin plots, inserted within the article and accompanied by a basic explanation (see caption Fig. 3), can be worth the extra complication, but it is more difficult to add such explanations in the dashboard. In contrast, many users may already know how to interpret box plots.

An additional problem with the violin plots in the dashboard is the representation of categories with very few points, where violin plots behave even worse than box plots; these combinations of categories can be avoided in a manuscript but not easily in a dashboard.

We consider that it would be even worse to duplicate the box plot tab and include both boxplots and violin plots, since it would complicate the interface for a very small benefit to most end users. That is the reason we decided to use only boxplots for the dashboard.

Minor

- I'd be wary of the sentence "The consideration of other processes in the fit, such as the Raman process and Quantum Tunneling of the Magnetisation (QTM), in principle results in more accurate values", which can be interpreted as a recommendation to always use more fitting parameters. Given that the literature is already filled with examples of overparametrisation, a clarification that this approach is to be taken only if one has enough data points in each of the regimes would be appreciated.

We agree with the reviewer on this point, a full fit should only be considered when the dataset includes enough points to avoid overparameterization.

We have modified the text accordingly (Section "A brief history of SIMs", page 2) and now reads:

"Given sufficient experimental information, considering other processes in the fit, such as the Raman process and Quantum Tunnelling of the Magnetisation (QTM), in principle results in more accurate values, which are denoted as $U_{\text{eff,ff}}$, $T_{0,\text{ff}}$."

- I still find the claim "The best metric for slow relaxation is hysteresis temperature" problematic. As discussed, this depends on the sweeping rate, which is by no means standardised. I suggest that it would be less controversial to convey the message that open hysteresis is a necessary condition for SMMs properties.

We thank the referee for this clarification. We have now added in section "A brief history of SIMs", page 2 :

"This translates in open hysteresis as a necessary condition for achieving molecular magnets."

- It is surprising that the authors do not reference Neese's work (Faraday Discuss., 2011,148, 229-238) when discussing "to improve the maximum Thyst is to maximise the total

effective spin (S), rather than the magnetic anisotropy (D).”, which demonstrates that S and D are not independent.

We thank the reviewer for his suggestion. The reference is now included as reference number 6 in the main text, in Section “A brief history of SIMs”, page 2:

“Initial models based on effective spin Hamiltonians gave rise to the relation $U_{\text{eff}} = DS_z^2$ and concluded that the best strategy to raise U_{eff} and, therefore, to improve the maximum T_{hyst} is to maximise the total effective spin (S), rather than the magnetic anisotropy (D).⁶”

-Fig 1a suggests that applying an external magnetic field should always be done as it leads to a maximum in the ac. I presume the authors just wanted to represent that in-field measurements remove QTM, but this is not at all obvious from reading the caption. Also, I would argue that a system that only shows slow relaxation of magnetisation in-field is not really a SMM. I suggest this figure or its explanation to be improved. In fig1b, the different elements of the fitting function should be aligned. Fig 1d is not mentioned in the main text, so one wonders what purpose it serves.

We thank the referee for this raising out this point, we have included an extra explanation in Fig. 1a caption, now it reads:

“being T_{B3H} the temperature for the maximum χ ” at 10^3 Hz in the presence of a magnetic field.”

In Fig. 1b, the elements of the fitting function are now aligned.

Fig. 1d is now mentioned in the main text (Section “A dataset and interactive dashboard for lanthanide SIMs”, page 6):

“At the same time, U_{eff} is rightfully criticised as an oversimplification that overlooks physically independent mechanisms (notably, Raman) that could be dominating the behaviour (Fig. 1d). Our dataset aimed to answer these questions.”

- I’d argue that the concept “ac magnetometry” is incorrect and suggest to use “ac susceptometry” instead.

We agree with the referee on this point. All the references to ‘ac magnetometry’ have been reformulated to “ac susceptometry” (in Section “A brief history of SIMs” and Results Section “A dataset and interactive dashboard for lanthanide SIMs”).

- The sentences “according to a simplified two-phonon Orbach model” and “An approximate relation between τ_0 and U_{eff} can be derived for two-phonon Orbach process within a Debye phonon model” need to be referenced.

We thank the referee for raising up this point. A citation to the classical text by Abragam and Bleaney (1970, Electron Paramagnetic Resonance) is now added to both sentences (Sections in Results “A dataset and interactive dashboard for lanthanide SIMs”, page 6, and “Orbach mechanism: oversimplified, predictive... a function of vibronic coupling?”, page 9). Reference number is 35.

- With respect to section 5.5 in the supplementary material, the derivation is much clearer now. As stated by the authors, Eq. 1 models a two-phonon Orbach process with a Debye model for the description of the phonons. Equating this to the generally employed Eq. 2 implies collapsing all the multiple-phonon processes encapsulated in U_{eff} to two-phonon processes. I suggest the authors comment on this and on the possibility that deviations from an exponent of 3 in Eq 5 might arise from the fact that what governs the overall reversal of magnetisation is not entirely due to two-phonon processes. Also, more of a question than a comment, does equating these two equations imply that the relaxation path occurs through the 2nd excited Kramers doublet?

We agree with the reviewer on the importance of showing the limitations of the two-phonon Orbach model. As the referee points out, equating eq. 1 to eq. 2 assumes that all higher-order phononic processes included in U_{eff} are reduced to a two-phonon process. Although maybe limited, this approach is meaningful and reproduces with a notably fidelity to the expected $n=3$ value. Of course, deviations from this model could be affected by that assumption including other non-Raman processes.

To answer the question from the reviewer we refer again to Abragam and Bleaney page 562, where a definition of the process is given:

“A second-order Raman process in which one phonon causes a virtual transition from one of the ground states to an excited state $|c\rangle$, followed by another virtual transition induced by the second phonon in which the magnetic ion returns from $|c\rangle$ to the other ground state.”

In this aspect, the relaxation path occurs through an excited kramers doublet not necessarily the 2nd but any with the correct phonon energy that permits the transition.

Also, we now clarify in Supplementary Information Section 5.5 Dependence of τ_0 vs U_{eff}

:

“Thus, U_{eff} in our dataset, and in equations (1) and (2), corresponds to Δ in the book.”

- It's odd to find copy pasted comments made by the referees in the main text: “historically, the fit to the Orbach process has been applied to the relaxation times obtained at the highest temperatures, even in systems where at very low temperatures the relaxation times were indicating purely quantum tunnelling or Raman relaxation mechanism.”

We apologise; we now rephrased the sentence. We sometimes have done this to exactly reflect some of the points raised by the referees, with which we agreed. It now reads in Supplementary Section 5.4 “The question of U_{eff} vs U_{eff} ”, page 54:

“Our observation could be due to the fact that, very often and especially in older works, a fit considering only the Orbach process was applied to the relaxation times obtained at the highest temperatures. This was done even in systems where at very low temperatures the magnetic behaviour points towards a purely quantum tunnelling or Raman relaxation mechanism”

- In the violin plots, why are the upper and lower adjacent values not shown? Is the median represented by the middle horizontal line?

We do not think that it is crucial to include the upper and lower adjacent values (nor the upper and lower inner fences) in the graph, since the information they provide does not

compensate for the slight complication of the interpretation for non-experts, which would require an additional definition of these metrics in the caption.

Indeed, the median is represented by the middle horizontal line. This is implicit in the caption, when we state that “the horizontal stripes mark the quartiles”, since the second quartile (Q2) is by definition the median of a dataset.

- The two statements in the sentence “Our analysis showed that the Arrhenius energy barrier correlates unexpectedly well with the magnetic memory, as both Orbach and Raman processes can be controlled by vibronic coupling.” feel disconnected, as there is no clear logical link between them.

We thank the reviewer for pointing out the errors in these sentences. We have rewritten the abstract and now reads:

“Our analysis showed that the Arrhenius energy barrier correlates unexpectedly well with the magnetic memory. Furthermore, as both Orbach and Raman processes can be controlled by vibronic coupling, chemical design of the coordination scheme may be used to reduce the relaxation rates. Indeed, only bis-phthalocyaninato sandwiches and metallocenes, with rigid ligands, consistently present magnetic memory up to high temperature.”

-One of the exciting prospects of having tools like this is to update them with new information to validate or challenge previous assumptions and conclusions. In that sense, I have the impression that this data set would benefit enormously from future phonon calculations, at least on the best-performing SMMs, especially to back claims made about spin-phonon coupling. Is this something the authors would consider to discuss?

As the reviewer punctualizes, an updated open database which includes spin-phonon calculations would be a reference tool in the field of SIMs and SpinQubits. We would in principle be happy to include these data into the dataset as soon as possible.

- I'd suggest that when selecting specific categories to represent in the app, only those appear labelled in the plot. For instance, in Supplementary Figure 11, only the Dy and Tb labels should appear.

We will take this suggestion, among other ideas, into consideration for future versions of the dashboard.

- I find the question “to determine whether one would need to employ $U_{\text{eff,ff}}$ instead.” ill-defined, as it is often the case that the relaxation profile only contains data on the Orbach mechanism, so accounting for the Raman and QTM parameters that would yield $U_{\text{eff,ff}}$ is just not possible. Maybe the authors can clarify this.

We rephrased for clarity, and now the sentence reads:

“We strived to quantify up to what level the value of U_{eff} is well correlated with the slow relaxation of the magnetisation. Ultimately, we wanted to verify whether a true correlation with magnetic relaxation is only possible when available data allows the fitting of $U_{\text{eff,ff}}$.”

- The sentence “the case of τ QTM (see Supplementary Figure 35).” Should read as “the case of τ QTM (see Supplementary Figure 36).”

We thank the referee for pointing this out. The figure number has been updated. Now the sentence reads: “Supporting this interpretation is also the fact that this correlation with $U_{\text{eff,ff}}$ is apparently absent in the case of τ_{QTM} (see Supplementary Fig. 36).”

Annex: new Supplementary Information Section

9.3 Linear relation between $\log(U_{\text{eff,ff}})$ vs $\log(C)$

Since there is a seeming relation between $U_{\text{eff,ff}}$ and C when represented in log-log plots (see Figure 5 in the main text and Supplementary Figure 34), we need to verify this more thoroughly by employing statistical analysis. We show the result of the analysis in this section, and the corresponding work for $U_{\text{eff,ff}}$ vs n and $U_{\text{eff,ff}}$ vs τ_{QTM} in sections 9.4 and 9.5 respectively. Since the apparent correlation was observed in the log-log plots, we have applied a logarithmic transformation on the two variables (\log_{10}).

The linear correlation and linear regression are statistical methods that study the linear relationship between two variables:

- The correlation quantifies how closely related two variables are. There are different correlation coefficients depending on the type of data we are working with. We will use Pearson's correlation (quantitative variables with a normal distribution, although it is quite robust to non-normality). In addition to calculating the correlation coefficient, we must also calculate its significance to accept whether or not there is a correlation between the variables. Finally, we will look at the size of the associated effect, which is what we know as the coefficient of determination R^2 . It is interpreted as the amount of variance of the dependent variable explained by the independent variable. It is obtained by squaring the correlation coefficient.

- A linear regression consists of generating a model that, based on the relationship between the two variables, allows predicting the value of one from the other.

In both cases we will check the applicability conditions. We will analyse whether both variables are correlated, and if they are, we will set up the regression model.

We created sub-databases because, depending on the variable we are working with, it may have more or less missing (NA) values. We have to eliminate them in order to be able to work.

We make the graphical representation of the 46 data that we have in this case (we have eliminated one record since when applying the logarithmic transformation on zero it is infinite). For this we have used the function `ggpairs` of the R package `GGally`.⁷⁰

Supplementary Figure 40 | $\text{Log}(U_{\text{eff_ff}})$ vs $\text{log}(C)$. From top left to bottom right: histogram with natural frequencies of $U_{\text{eff_ff}}$ values, numerical value for the correlation, dotplot including the linear fit and histogram with natural frequencies of C .

We can see that the behaviour of the variables is very similar to a normal and that the calculated correlation coefficient is significant with a value of -0.729, which indicates that the association between the variables is high. In addition, the sign is negative, which indicates that when the values of $U_{\text{eff_ff}}$ increase, the values of C decrease.

The Pearson's test result is as follows:

```
##
## Pearson's product-moment correlation
##
## data: datos1_C$logUeff_ff[-6] and datos1_C$logC[-6]
## t = -6.9775, df = 43, p-value = 1.387e-08
## alternative hypothesis: true correlation is not equal to 0
## 95 percent confidence interval:
## -0.8421097 -0.5535775
## sample estimates:
## cor
## -0.7287019
```

The coefficient of determination associated with the correlation coefficient is:

```
## [1] 0.5310065
```

Next we run the linear model. We obtain the following output:

```
## Estimate Std. Error Lower 95% Upper 95% P-value
## (Intercept) 4.251 0.922 2.392 6.11 <0.001

## logUeff_ff -3.038 0.435 -3.916 -2.16 <0.001
## R Squared 0.531
## Adj.R Squared 0.5201
```

We see that the estimate obtained for U_{eff_ff} is -3.038 and that it is significant. This means that: a one-unit increase in the predictor $\log(U_{eff_ff})$ causes an average change of -3.038 units in the response variable $\log(C)$ while keeping all the other predictors constant (in this case we have no more predictors). In the same way as with the calculation of the correlation coefficient, the relationship is inverse.

Let us check the applicability conditions. From the model we obtain the following graphs:

Supplementary Figure 41 | Applicability conditions for the relation between $\log(U_{eff_ff})$ vs $\log(C)$. From top left to bottom right: residuals vs fitted values, standardised residuals vs their theoretical values, square root of the standardised residuals vs the fitted values and standardised residuals vs leverage.

Visually, it seems that the condition of linearity and normality of the residuals is fulfilled, but let us check them all:

- **Linearity:**

Supplementary Figure 42 | Linearity condition of the model. Value of the residual vs the prediction of the model.

The residuals are randomly distributed around 0 so linearity is accepted.

- **Normality:**

Supplementary Figure 43 | Normality condition of the model. Frequency density vs residuals.

Let's carry out the corresponding test to check it:

```
##
## Shapiro-Wilk normality test
##
## data: mod_C1$residuals
## W = 0.98134, p-value = 0.6743
```

The p-value obtained is not significant, so we are not in a position to reject the normality hypothesis, so the normal distribution of the residuals is confirmed.

- **Homogeneity of variance:**

We performed the Breusch-Pagan test to check for this:

```
##
## studentized Breusch-Pagan test
##
## data: mod_C1
## BP = 3.6016, df = 1, p-value = 0.05772
```

The p-value obtained is not significant, so we accept the condition of homogeneity of variances.

- **Autocorrelation of residuals:**

Supplementary Figure 44 | Autocorrelation of the residuals. Values of the residuals vs their index.

The representation of the residuals does not show any trend.

● **Outliers:**

Supplementary Figure 45 | Outlier in $\log(U_{\text{eff,ff}})$ vs $\log(C)$. Scatter diagram. Outliers are marked in red.

We have identified the outliers and recalculated the least squares linear regression. The plot shows that there is practically no change with or without the outliers.

In conclusion, all conditions to apply a least squares linear regression model are met and that the obtained p-value indicates a significant association for both variables. The adjusted R^2 is 52%.

9.4 Linear relation between $\log(U_{\text{eff,ff}})$ vs $\log(n)$

We make the graphical representation of the 50 data in this case. We have used, as in the previous section, the function `ggpairs` of the R package `GGally`.⁷⁰

Supplementary Figure 46 | $\text{Log}(U_{\text{eff,ff}})$ vs $\text{log}(n)$. From top left to bottom right: histogram with natural frequencies of $U_{\text{eff,ff}}$ values, numerical value for the correlation, dotplot including the linear fit and histogram with natural frequencies of n .

It is shown that the behaviour of the variables is very similar to a normal distribution and that the calculated correlation coefficient is significant with a value of -0.481. This indicates that the association between the variables is moderate. Furthermore, the sign is negative, which means that when the values of $U_{\text{eff,ff}}$ increase, consequently the values of n decrease.

The Pearson's test is as follows:

```
##  
## Pearson's product-moment correlation  
##  
## data: datos1_n$logUeff_ff and datos1_n$logn  
## t = -3.8043, df = 48, p-value = 0.0004023  
## alternative hypothesis: true correlation is not equal to 0  
## 95 percent confidence interval:  
## -0.6699142 -0.2343687  
## sample estimates:  
## cor  
## -0.4813173
```

The coefficient of determination associated with the correlation coefficient is:

```
## [1] 0.2316663
```

Next we run the linear model. We obtain the following output:

```
## Estimate Std. Error Lower 95% Upper 95% P-value
## (Intercept) 1.001 0.084 0.831 1.17 <0.001

## logUeff_ff -0.15 0.039 -0.229 -0.071 <0.001
## R Squared 0.2317
## Adj.R Squared 0.2157
```

We can see that the estimate obtained for $\log(U_{\text{eff_ff}})$ is -0.15 and that it is significant. This indicates that an increase of one-unit in the predictor ($\log(U_{\text{eff_ff}})$) causes an average change of -0.15 units in the response variable (in this case is $\log(n)$) while keeping all the other predictors constant (in this case we have no more predictors). Besides as with the calculation of the correlation coefficient, the relationship is inverse.

Let us check the applicability conditions. From the model we obtain the following graphs:

Supplementary Figure 47 | Applicability conditions for the relation between $\log(U_{\text{eff_ff}})$ vs $\log(n)$. From top left to bottom right: residuals vs fitted values, standardised residuals vs their theoretical values, square root of the standardised residuals vs the fitted values and standardised residuals vs leverage.

- **Linearity:**

Supplementary Figure 48 | Linearity condition of the model. Value of the residual vs the prediction of the model.

The residuals are randomly distributed around 0 so linearity is accepted.

- **Normality:**

Supplementary Figure 49 | Normality condition of the model. Frequency density vs residuals.

Then we carry out the corresponding test:

```
##
## Shapiro-Wilk normality test
##
## data: mod_n1$residuals
## W = 0.97245, p-value = 0.2904
```

The p-value obtained is not significant, so we are not in a position to reject the normality hypothesis, so the normal distribution of the residuals is confirmed.

- **Homogeneity of variance:**

We performed the Breush-Pagan test to check for this:

```
##
## studentized Breusch-Pagan test
##
## data: mod_n1
## BP = 3.3668, df = 1, p-value = 0.06652
```

The p-value obtained is non-significant, so we accept the condition of homogeneity of variances.

- **Autocorrelation of residuals:**

Supplementary Figure 50 | Autocorrelation of the residuals. Values of the residuals vs their index.

The representation of the residuals does not show any trend.

- **Outliers:**

Supplementary Figure 51 | Outlier in $\log(n)$ vs $\log(U_{\text{eff,ff}})$. Scatter diagram. Outliers are marked in red

We have identified the outliers and recalculated the least squares linear regression. A change in the slope can be seen. Furthermore, if we remove the outliers the calculated estimation changes from -0.15 to -0.09.

In conclusion, all conditions to apply a least squares linear regression model are met, and the obtained p-value indicates a significant association for both variables. The adjusted R^2 is 21.5%.

9.5 Linear relation between $\log(U_{\text{eff,ff}})$ vs $\log(\tau_{\text{QTM}})$

Next, we make the graphical representation of the 58 entries with values of τ_{QTM} . We have used, as in the previous section, the function `ggpairs` of the R package `GGally`.⁷⁰

Supplementary Figure 52 | $\text{Log}(U_{\text{eff,ff}})$ vs $\text{log}(\tau_{\text{QTM}})$. From top left to bottom right: histogram with natural frequencies of $U_{\text{eff,ff}}$ values, numerical value for the correlation, dotplot including the linear fit and histogram with natural frequencies of n .

We can see that the behaviour of the variables is very similar to a normal distribution and that the calculated correlation coefficient is not significant and small. This means either there is not an association between the variables or it is very small.

The Pearson's test is as follows:

```
##  
## Pearson's product-moment correlation  
##  
## data: datos1_tauqtm$logUeff_ff and datos1_tauqtm$logtau_qtm  
## t = -0.68349, df = 37, p-value = 0.4986  
## alternative hypothesis: true correlation is not equal to 0  
## 95 percent confidence interval:  
## -0.4126421 0.2112983  
## sample estimates:  
## cor  
## -0.1116631
```

We can see that the association is non-significant so that it makes no sense to test a linear model with the variables.

Anyhow, we show the result of the model:

```
## Estimate Std. Error Lower 95% Upper 95% P-value
## (Intercept) -0.634 1.777 -4.234 2.967 0.723
## logUeff_ff -0.62 0.907 -2.459 1.218 0.499
## R Squared 0.0125
## Adj.R Squared -0.0142
```

We can see that the calculated estimate for Ueff_ff is non-significant.

REVIEWER COMMENTS

Reviewer #3 (Remarks to the Author):

The authors have successfully addressed all my comments and as such I recommend the study for publication in Nature Communications. There are only two minor remarks I would like to make:

- The authors state that “Orbach and Raman processes can be controlled by vibronic coupling” and “this can be related to the fact that Ueff, C are heavily controlled by the spin-vibrational relaxation”. I’d argue that “controlled” is too strong of a word as it implies a prior knowledge on how to achieve that control consistently and effectively, which is currently lacking in the field. I would the change “controlled” by “affected”.

- The paper is missing a key reference on spin-phonon coupling (J. Am. Chem. Soc. 2021, 143, 34, 13633–13645). I think that framing the obtained correlations in terms of the insights provided in that paper would improve the study.

Answer to the referees (final revisions)

REVIEWERS' COMMENTS

Reviewer #3 (Remarks to the Author):

The authors have successfully addressed all my comments and as such I recommend the study for publication in Nature Communications. There are only two minor remarks I would like to make:

- The authors state that “Orbach and Raman processes can be controlled by vibronic coupling” and “this can be related to the fact that Ueff, C are heavily controlled by the spin-vibrational relaxation”. I’d argue that “controlled” is too strong of a word as it implies a prior knowledge on how to achieve that control consistently and effectively, which is currently lacking in the field. I would the change “controlled” by “affected”.

We thank the reviewer for the good appreciation of our work and her/his suggestions to improve the final version. We agree that “affected” is more realistic and changed both instances of “controlled” to “affected”.

- The paper is missing a key reference on spin-phonon coupling (J. Am. Chem. Soc. 2021, 143, 34, 13633-13645). I think that framing the obtained correlations in terms of the insights provided in that paper would improve the study.

We agree and added this reference.